## Registered report

cognition/psychology

bilingual advantage, cognitive control, inhibitory control, infant development

**Author for correspondence:**
Dean D'Souza
e-mail: dean.dsouza@aru.ac.uk

# Is mere exposure enough? The effects of bilingual environments on infant cognitive development

Dean D'Souza[1], Daniel Brady[2], Jennifer X. Haensel[3,4] and Hana D'Souza[5]

[1]Faculty of Science and Engineering, Anglia Ruskin University, Cambridge, UK
[2]School of Psychology and Clinical Language Sciences, University of Reading, Reading, UK
[3]Department of Computer Science, University of Bath, Bath, UK
[4]Centre for Brain and Cognitive Development, Birkbeck, University of London, London, UK
[5]Department of Psychology and Newnham College, University of Cambridge, Cambridge, UK

 DD, 0000-0002-3053-2484

Bilinguals purportedly outperform monolinguals in non-verbal tasks of cognitive control (the 'bilingual advantage'). The most common explanation is that managing two languages during language production constantly draws upon, and thus strengthens, domain-general inhibitory mechanisms (Green 1998 *Biling. Lang. Cogn.* **1**, 67–81. (doi:10.1017/S1366728998000133)). However, this theory cannot explain why a bilingual advantage has been found in preverbal infants (Kovacs & Mehler 2009 *Proc. Natl Acad. Sci. USA* **106**, 6556–6560. (doi:10.1073/pnas.0811323106)). An alternative explanation is needed. We propose that exposure to more varied, less predictable (language) environments drive infants to sample more by placing less weight on consolidating familiar information in order to orient sooner to (and explore) new stimuli. To confirm the bilingual advantage in infants and test our proposal, we administered four gaze-contingent eye-tracking tasks to seven- to nine-month-old infants who were being raised in either bilingual ($n = 51$) or monolingual ($n = 51$) homes. We could not replicate the finding by Kovacs and Mehler that bilingual but not monolingual infants inhibit learned behaviour (experiment 1). However, we found that infants exposed to bilingual environments do indeed explore more than those exposed to monolingual environments, by potentially disengaging attention faster from one stimulus in order to shift attention to another (experiment 3) and by switching attention more frequently between stimuli (experiment 4). These data suggest that experience-driven adaptations may indeed result in infants exposed to bilingual environments switching attention more frequently than infants exposed to a monolingual environment.

# 1. Introduction

Bilinguals often outperform monolinguals in non-verbal tasks of cognitive control. For example, a meta-analysis of studies that compared the performance of bilinguals and monolinguals on conflict resolution tasks (e.g. the Stroop task) revealed a moderately significant bilingual advantage [1]. At first blush, this finding augments our understanding of far-transfer effects (how practice in one domain (language) results in changes to other (non-language) domains) and may inform educational policies and social practice. However, close inspection of the empirical data yield inconclusive results. For example, the effect reported in the above meta-analysis appears to be driven by data from a single study. Furthermore, the meta-analysis also revealed a large and significant main effect of 'laboratory' (research group). The inconsistent results, combined with publication bias [2], have led many scientists (e.g. [3–5]) to question whether the bilingual advantage is real or merely an artefact of particular research practices. Moreover, significant bilingual advantages have not been observed in studies with large sample sizes (see [5] for discussion). To progress beyond the controversy and advance the science, in addition to carrying out more studies, we must come up with a theory that can account for the inconsistencies in the literature and explain *when*, *how* and *why* learning two or more languages improves cognitive control. This is the focus of the current paper.

The most influential explanation for the bilingual advantage is Green's [6] proposal that managing two or more languages during language production constantly draws upon, and thus strengthens, *domain-general* cognitive control processes that select words in the intended language while inhibiting the activation of words in the unintended language. Neurophysiological evidence in support of Green's [6] inhibitory control model came from a study by Blanco-Elorrieta & Pylkkanen [7], which demonstrated that whereas switching languages in comprehension draws upon language-specific control processes (in anterior cingulate cortex), switching languages in production recruits domain-general cognitive control processes (in dorsolateral prefrontal cortex). However, as D'Souza & D'Souza [8] pointed out, a more direct test of the inhibitory control model would have probed for a bilingual advantage in participants who can comprehend but not produce language—i.e. in 6–11-month-old 'preverbal' infants. According to the inhibitory control model, preverbal infants raised in bilingual homes should not show a bilingual advantage. Yet studies suggest that they do [9,10]. For example, in an eye-tracking study of seven-month-old infants, Kovacs & Mehler [9] found that all infants could respond to a speech or visual cue to anticipate a reward on one side of a screen, but only infants raised in bilingual homes could inhibit their learned response and redirect their anticipatory looks when the cue began signalling the reward on the opposite side. Kovacs & Mehler [9] concluded that processing representations from two or more languages somehow increases the infant's ability to learn a new response and suppress their old one. The inhibitory control model can neither predict nor explain these data, because seven-month-old infants can only comprehend, not produce, words [11].[1]

Why do some studies suggest there is a bilingual advantage in preverbal infants? We have argued that mere exposure to bilingual environments may lead to experience-driven adaptations that confer both cognitive advantages and disadvantages [8,13]. But what adaptations, how, why and when?

## 1.1. Adapting to variable environments

All biological systems, including human infants, are driven to minimize uncertainty, the difference between the infant's predictions about its sensory inputs (embodied in its models of the world) and the sensations it encounters [14]. This general adaptive drive helps the infant to better model and anticipate events in its ever-changing world [14]. The infant minimizes uncertainty by sampling, selecting and acting on the external world (resulting in different sensory input [13]) and by altering its models and predictions (often resulting in different perceptions [15]). In other words, development involves an active process of calibration, whereby the infant learns to sample its econiche and predict events by modelling its interactions within the econiche. The more variable the environment, however, the more exploration (sampling) is required to minimize uncertainty and generate more confident estimates (see [13] for discussion). Consequently, a child's behaviour, model and predictions are yoked to its econiche. Exposure to different environments will result in different models and predictions.

[1]Green's latest proposal—the adaptive control hypothesis [12]—complements his inhibitory control model by suggesting that the bilingual advantage arises as domain-general control processes involved in managing two or more languages are altered by the recurrent demands placed upon them by the interactional context (single language use, dual language use, dense code-switching). However, like Green [6], this proposal also relies on speech production and thus neither predicts nor explains the infant data.

One variable in the infant's environment is the number of languages that the child regularly hears. Infants who regularly hear two or more languages are necessarily exposed to more varied and less predictable language input than infants who regularly hear only one language. These 'bilingual' infants are also likely to receive less input from each language than 'monolingual' infants from their one language. Because some bilingual mothers make more errors than monolingual mothers [16], bilingual infants may also receive less accurate input. Given these exogenous sampling constraints, how does the bilingual infant keep pace (developmentally) with its monolingual peers? How does the bilingual infant minimize uncertainty to the same extent as the monolingual infant when it is exposed to a more variable (language) environment and receives fewer samples? As mentioned above, it can minimize uncertainty by acting (switching attention) and altering its models and predictions.

According to Mareschal *et al*. [17], infants acquire and develop multiple, partial representations that are just sufficient for 'on-the-fly processing'. D'Souza & D'Souza [8] hypothesized that bilingual infants adapt to their more varied, less predictable and less accurate language environments by learning to construct, and get by on, more partial representations *or less detailed models of their environment*, which would enable them to redirect their attention sooner to less familiar (but equally important) stimuli (e.g. their second language).[2] In other words, whereas monolingual infants are drawn to familiar stimuli so they can build detailed representations of their environment and reduce uncertainty, bilingual infants *err on the side of exploration* and place more weight on novel stimuli. That is, to collect more samples from their more varied environments, bilinguals place less weight on familiar information. If this is the case, then we would expect bilingual infants to show less familiarity preference (which is something that helps infants to build more detailed models) and more novelty preference than monolingual controls. Indeed, Singh *et al*. [10] found that six-month-old bilinguals look increasingly less at a repeatedly presented visual stimulus than monolinguals. Moreover, if bilinguals learn to get by on less complete internal models, it would explain why Folke *et al*. [18] found that bilingual adults show a disadvantage in metacognition, the ability to evaluate one's own cognitive processes. Bilinguals would find it more difficult to evaluate their own cognitive processes because—according to our hypothesis—they are getting by on more fragmentary models of the external world.

Why would adaptations to variation in one domain (auditory-verbal) affect other domains (e.g. visual)? Building models of the external world might be a domain-general process, because it often involves the integration of action (e.g. shifting visual attention), perception and multisensory information processing (e.g. matching sounds to lip movements).

## 1.2. Predictions

We hypothesized that exposure to more varied language environments drive infants to explore (sample) further by constructing less detailed models of their environments and placing more weight on novel information.[3] Getting by on less detailed models would allow the child to switch faster to novel stimuli and thus sample more from their environments. We tested our hypothesis by running a series of experiments that sought to replicate with a larger sample—and thus support—Kovacs & Mehler's [9] finding of a bilingual advantage in infants (experiment 1) and probe whether bilingual infants build, and get by on, less detailed representations of visual stimuli (experiment 2), shift attention faster to a second visual stimulus (experiment 3), and are (thus) less sensitive to the minute detail of a visual stimulus (experiment 4).

Specifically, we predicted that bilingual infants would: (i) be better at inhibiting a learned behaviour (experiment 1); (ii) respond more appropriately to more fragmented—or *less detailed*—visual stimuli (experiment 2); (iii) be more likely to abandon the visual processing of a stimulus to shift attention to a novel stimulus (experiment 3); and (iv) switch more frequently between two visual stimuli, spend less time visually processing a familiar stimulus, and thus be worse at remembering the details of a visual stimulus (experiment 4; table 1). Although these infant studies can neither support nor disprove the theory that a bilingual advantage results from managing two or more languages *during production* in older children and adults, if we observe a significant group difference we could argue that it

---

[2]Models and representations are similar concepts. A *representation* is an information state in the brain, expressed through patterns of neural activity that (to varying extents) reflect states in the world and contribute to adaptive behaviour. A *model* is a representation of a selected part of the world (the 'target system'). In this paper, we use the words *model* and *representations* interchangeably.

[3]More exploration—at the expense of building more detailed representations of familiar stimuli—may also result in prolonged neural plasticity and functional specialization [13].

**Table 1.** Hypotheses our experiments tested.

| hypotheses: compared to monolinguals, bilinguals will… | experiment no. |
| --- | --- |
| be better at inhibiting a learned behaviour | 1 |
| respond more appropriately to more fragmented—or *less detailed*—visual stimuli | 2 |
| be more likely to abandon the visual processing of a stimulus to shift attention to a novel stimulus | 3 |
| switch more frequently between two visual stimuli | 4 |
| spend less time visually processing a familiar stimulus | 4 |
| be worse at remembering the details of a visual stimulus | 4 |

reflects experience-dependent adaptations that occur because of regular *exposure* to two or more languages. If we fail to observe a significant group difference, we could argue that there is currently insufficient evidence to conclude there is a bilingual advantage in *preverbal infants*.

We focused on the visual domain, because of its relevance to the bilingual advantage and far-transfer effects (i.e. whether training in the language domain transfers to non-language domains). The present study is important because if these adaptations do indeed exist, they are likely to constrain learning and development across multiple neurocognitive domains. This would have immediate implications for theory, and may impact social practices (e.g. parenting) and education policies.

# 2. Material and methods

## 2.1. Participants

We collected data from 102 infants (seven to nine months of age), of whom 51 were raised in 'bilingual' homes and 51 in 'monolingual' homes.[4] In line with previous research (e.g. [10,21]), infants were considered 'bilingual' if they had daily exposure to two or more languages and heard their first language no more than 75% of the time; they were considered 'monolingual' if exposed to their first language for at least 90% of the time (measured using the Language Exposure Questionnaire (LEQ)— see §2.2.5). Infants born between 38 and 42 weeks of gestational age were included in this study—as well as preterm infants born between 36 and 37 weeks if they weighed over 2.38 kg (5 lbs 4 oz) at birth. Families on our database received a parent report questionnaire (see §2.2.5). We used the database and questionnaire data to recruit two groups (bilingual and monolingual) that were closely matched on age, gender and parents' socioeconomic status (SES). The SES score was a composite of four weighted scores based on the carers' (1) postcode (as an index of socioeconomic deprivation), (2) education attainment, (3) household income and (4) occupation. For details, see the electronic supplementary material. We checked that the two groups did not significantly differ from each other on age, gender, or parents' SES. The age of the bilingual infants ($M = 254$ days, s.d. $= 26$) was not significantly different from the age of the monolingual infants ($M = 260$ days, s.d. $= 27$), $t_{145} = 1.36$, $p = 0.175$. There was no significant association between group and gender (64% of the bilingual infants were male versus 49% of the monolingual infants), $\chi_2^2 = 4.37$, $p = 0.112$. Finally, the SES of the bilingual families ($M = 0.66$, s.d. $= 0.16$) was not significantly different from the SES of the monolingual families ($M = 0.64$, s.d. $= 0.14$), $t_{145} = 0.73$, $p = 0.466$.

Participants were recruited and tested until, for each task, we had useable data from 51 bilingual and 51 monolingual infants. We defined useable data as eye-tracking data (gaze patterns) from at least 75% of the trials in the task. So, if an infant provided eye-tracking data (gaze patterns) for 75% of the trials in experiment 1, but less than 75% of the trials in experiment 2, then the data the infant provided for experiment 1 was analysed, but not the data for experiment 2, and an extra participant was recruited. Furthermore, infants had to provide useable data for at least one of the two test trials in experiment 2.

[4]Kovacs & Mehler [9] found group differences with sample sizes of just 20 per group. However, because a true effect is likely to be smaller than that indicated by an initial study (see [19] for discussion), we opted for a larger sample size. In the interests of informing public policy, we sought to detect medium-sized effects ($d = 0.50$; power $= 0.80$, $\alpha = 0.05$) rather than small-sized effects. For a medium-sized effect of 0.50, the probability that you can guess which group a person is in from their test score is 60% [20]. Where possible, required sample sizes were calculated using power analyses (see below for details).

Extra participants were recruited only towards the end of the study and partook in as many experiments as needed. Thus, if towards the end of the study we had useable data for experiments 1–4 from 50, 48, 49 and 50 bilinguals, respectively, then the first extra bilingual participant had to do all four experiments, the second had to do experiments 2 and 3 only, and the third had to do experiment 2 only. For each experimental task, we ran an independent $t$-test to check that the two groups did not significantly differ on the number of valid trials that they provided.

## 2.2. Materials and procedure

Four eye-tracking experiments were carried out. Because we may want to look at individual differences, the order of experiments was fixed: the switch task, the visual memory task, the representations task and the gap-overlap task. All experiments involved a Tobii Pro TX300 remote eye tracker to capture moment-to-moment point of gaze at a sampling rate of 120 Hz, with measurement accuracy of 0.4° (screen size: 58.42 cm; aspect ratio: 16 : 9; screen resolution: 1920 × 1080). The tracking equipment and stimulus presentation were controlled using customized scripts in MATLAB R2013a. A camera mounted directly above the horizontal midpoint of the screen was used to monitor and record infant behaviour. Auditory stimuli were delivered via two speakers positioned behind the display monitor and facing the participant. The infant sat on their carer's lap, in a dimly lit featureless room, facing the stimulus-presentation screen with their eyes at approximately 65 cm from the screen. Carers were asked to close their eyes during the experiment. A 5-point calibration was used. This involved an attractive swirling shape, which moved across the screen and stopped at the centre and four corners. At each stop-point, the experimenter—who was observing the child's face and eye movements via a video camera attached to the eye tracker—manually re-started the ball moving when the child looked at it. If calibration was good for at least 3 of the 5 points (e.g. precision and accuracy within 1.5° and 5°, respectively; Tobii Technology AB, 2011), then the eye-tracking experiments began. If not, then further attempts at calibration were made unless, or until, the infant became too fussy to participate.

### 2.2.1. Experiment 1: the switch task

Experiment 1—a replication of Kovacs & Mehler [9] but with a much larger sample size and arguably more engaging stimuli—was designed to provide evidence in support of, or against, the claim that bilingual infants are significantly better than monolingual infants at learning a new response by suppressing an old one. Infants must learn, over nine 'pre-switch' trials, to make an anticipatory look to one side of a screen. They must subsequently—during nine 'post-switch' trials—suppress their learned response and instead make an anticipatory look to the other side of the screen. Infants succeed on this task if, during the 'post-switch' phase, they update their prediction and inhibit their first learned response—demonstrating the rudiments of cognitive control.

#### 2.2.1.1. Stimuli

The experimental stimuli were 18 visual cues, four visual-auditory attention grabbers and six visual-auditory rewards. The *cues* consisted of sequences of three simple geometric shapes, with identical shapes either at the beginning (forming an AAB structure) or end (ABB). Nine AAB sequences and nine ABB sequences were constructed from three 'A' shapes (arrow, circle, pentagon) and three 'B' shapes (star, triangle, moon). The shapes were 7.9 cm wide (300 pixels wide, which subtend at an angle of 7.0°) and different colours. The (four) attention grabbers and (six) rewards comprised 10 (5.3 cm/200 pixels wide, 4.7°) dynamic colourful pictures (e.g. a rotating flower) paired with one of six interesting sounds (e.g. 'beep!'). The rewards appeared within white 5.3 cm squares (4.7°) located 31.7 cm (1200 pixels, 27.4°) apart on the left or right side of a grey screen.

#### 2.2.1.2. Procedure

The experiment consisted of nine pre-switch trials, followed by nine post-switch trials. All trials comprised four consecutive displays: fixation, cue, anticipation, and reward (figure 1). All trials started with the *fixation* display: two white squares (on either side of a grey screen) and, in the middle of the screen, an *attention getter* (an interesting visual stimulus accompanied by an interesting sound). The white squares remained onscreen throughout the entire experiment. The fixation display lasted 0.5 s and was followed by the *cue* display: three colourful shapes were presented, one after the other, following one pattern (AAB or ABB) in the 'pre-switch' phase and a different pattern (ABB or AAB)

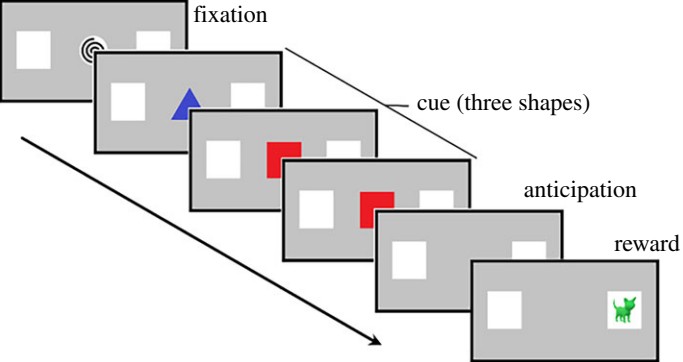

**Figure 1.** In the switch task (adapted from [9]), a fixation stimulus was presented for 0.5 s to attract the infant's attention. This was followed by the cue: three shapes that appeared one after the other. Each shape was onscreen for 0.8 s; the temporal gap between each shape was 0.3 s. An anticipation display (1 s) preceded presentation of the reward (2 s). The reward was always displayed on the same side of the screen during nine pre-switch trials and on the other side during nine post-switch trials.

in the 'post-switch' phase. Each shape remained onscreen for 0.8 s and the temporal gap between each shape was 0.3 s. At the offset of the visual cue, the *anticipatory* period began and lasted for 1 s; after which followed the onset of the *reward* display: a visual reward appeared inside one of the white boxes, accompanied with an auditory reward. The reward display lasted for 2 s. The reward was always displayed on the same side of the screen during the pre-switch trials and on the other side during the post-switch trials. The order of the cue type (AAB, ABB) and reward side (left, right) was counterbalanced (i.e. each child completed one of four presentation orders). The procedure lasted around 2 min.

### 2.2.1.3. Coding and analysis

Two areas-of-interest (AOI) were defined around the white squares that the rewards appeared in. The predefined AOIs extended 1 cm around the white squares in case the infant focused on the edge of the visual reward and to account for small deviations in the quality of the calibration. An 'anticipatory look' was defined as a saccade to one of the AOIs (left or right) that occurred within a 1 s time window starting 150 ms after cue offset and ending 150 ms after onset of the visual reward [9]. If, at any point during the 1 s time window, the infant performed an anticipatory look to the AOI where the reward would appear, the trial was automatically coded as 'correct'; an anticipatory look to the other AOI was automatically coded as 'incorrect'. If the infant looked at both AOIs (left and right), then the AOI of the longer look was coded [9]. All other onscreen looking behaviour was automatically coded as 'invalid' and discarded. If the infant looked away from the screen during the time window, the trial was automatically coded as 'invalid' and discarded.

If an infant has learnt that the cue predicts the location of the visual reward, then, over the trials, they should have increased their anticipatory looks to the region of the screen where they expected the reward to appear. Based on Kovacs & Mehler [9], we predicted that in the post-switch phase only the bilinguals would learn to correctly anticipate the reward. To test this hypothesis, we carried out two (pre-switch, post-switch) mixed-effects logistic regressions.[5] The outcome variable was *anticipatory look* (correct versus incorrect); the predictor variables were *trial* (1–9) and *group* (monolingual versus bilingual). Random coefficients and an autoregressive covariance structure were assumed. If we found a statistically significant effect of group, then we followed it up with two one-sample *t*-tests to ascertain whether the bilingual/monolingual infants anticipated the reward (proportion of correct anticipatory looks to total anticipatory looks, averaged over the last three trials) greater than expected by chance (0.50).[6] If they did, and if at least 50% of them had contributed anticipatory looks, then we can conclude that they had inhibited their first learned response and learned the new one.

---

[5]There does not appear to be a consensus on how to calculate *a priori* the required sample size for a mixed-effects logistic regression. However, simulation studies suggest that nine observations per individual (918 observations in total) should provide sufficient power (greater than 0.80; [22]).

[6]To detect a medium-sized effect ($d = 0.50$), we required a sample size of at least 27 per group (G*Power). Our sample size of 51 per group was sufficient to detect a small-medium effect ($d = 0.35$; when power = 0.8 and $\alpha = 0.05$; G*Power).

## 2.2.2. Experiment 2: the representations task

Experiment 2 probes whether bilingual infants build, and get by on, less detailed representations of visual stimuli than monolingual infants. Participants had to learn to respond to relatively detailed visual cues. Infants succeed on this task if, during the test phase, they respond appropriately to *less detailed* visual stimuli.

### 2.2.2.1. Stimuli

The stimuli were six visual cues, four visual-auditory attention grabbers and six visual-auditory rewards. The visual cues comprised three fragmented line drawings of an elephant (two 'training' cues and one 'test' cue) and three fragmented drawings of a snowman (two 'training' cues and one 'test' cue). Each drawing consisted of black fragmented lines on a white 7.9 cm wide (7.0°) square (figure 2). The attention grabbers and rewards comprised 10 dynamic colourful pictures (all 5.3 cm wide; 4.7°) paired with one of six interesting sounds. Like experiment 1, the rewards appeared within white 5.3 cm squares (4.7°) located 31.7 cm (27.4°) apart on the left or right side of a grey screen.

### 2.2.2.2. Procedure

The experiment consisted of 24 training trials, followed by two test trials. Like experiment 1, all trials comprised four consecutive displays: fixation, cue, anticipation, and reward. All trials started with the *fixation* display: two white squares (on either side of a grey screen) and, in the middle of the screen, an *attention getter* (an interesting visual stimulus accompanied by an interesting sound). The white squares remained onscreen throughout the entire experiment. The fixation display lasted 0.5 s and was followed by the *cue* display: a visual cue (either a fragmented elephant or a fragmented snowman) appeared at the centre of the screen for 2.5 s. At the offset of the visual cue, the *anticipatory* period began and lasted for 1 s; after which a visual reward appeared inside one of the white boxes, accompanied by an auditory reward. The reward display lasted for 2 s. Audio was played during the attention grabber and reward screens: four different ringing sounds for the attention grabber and six different sounds for the reward.

For half the bilingual/monolingual infants, the elephant cue signalled the appearance of a reward on the left side of the screen, while the snowman cue signalled the appearance of a reward on the right side. For the other half, the elephant cued a reward on the right side, while the snowman cued a reward on the left side. Critically, during the test phase (two test trials: one elephant, one snowman), participants were presented with an even more fragmented cue than during the training phase. During both training and test phases, cue order was randomized for each participant. The procedure lasted around 3 min.

### 2.2.2.3. Coding and analysis

Two AOI were defined around the white squares that the rewards appeared in. The predefined AOIs extended 1 cm around the white squares in case the infant focused on the edge of the visual reward and to account for small deviations in the quality of the calibration. An 'anticipatory look' was defined as a saccade to one of the AOIs (left or right) that occurred during a 1 s time window starting 150 ms after cue offset and ending 150 ms after reward onset. If, at any point during the 1 s time window, the infant performed an anticipatory look to the AOI where the reward would appear, the trial was automatically coded as 'correct'; an anticipatory look to the other AOI was automatically coded as 'incorrect'. If the infant looked at both AOIs (left and right), then the AOI of the longer look was coded. All other onscreen looking behaviour was automatically coded as 'invalid' and discarded. If the infant looked away from the screen during the time window, the trial was automatically coded as 'invalid' and discarded.

If an infant has learnt that the cue type (elephant, snowman) predicts the location of a visual reward, then, over the trials, they should have increased their anticipatory looks to the region of the screen where they expected the reward to appear. To check whether they had learned to associate a cue with the location of a reward, we carried out a mixed-effects logistic regression.[7] The outcome variable was *anticipatory look* (correct versus incorrect); the predictor variables were *trial* (1–24) and *group* (monolingual versus bilingual). Random coefficients and an autoregressive covariance structure were

---

[7]Simulation studies suggest that 24 observations per individual (2448 observations in total) should provide sufficient power (greater than 0.80 [22]).

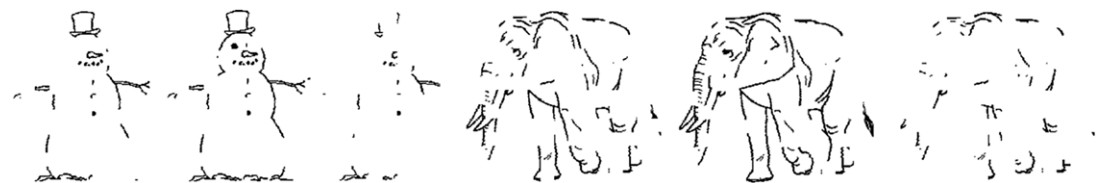

**Figure 2.** The two snowmen on the left are training stimuli (for the representations task). The snowman on the right is a test stimulus (i.e. a less-detailed representation of the snowmen on the left). The two leftmost elephants are also training stimuli (for the representations task). The elephant on the right is a test stimulus (i.e. a less-detailed representation of the elephants on the left).

assumed. If we found that correct anticipatory looks increased significantly across trials in both groups, then we followed up the result with two one-sample *t*-tests to ascertain whether the bilingual/monolingual infants correctly anticipated the reward (proportion of correct anticipatory looks to total anticipatory looks, averaged over the last three trials) greater than expected by chance (0.50).[8] If they did, and if at least 50% of them had contributed anticipatory looks, then we may conclude that they made an association between a cue and the location of a reward. In the test phase, however, we predicted that the bilinguals would be more likely to use the more-fragmented 'test-trial' cue to anticipate the side of the reward than the monolinguals. To test this hypothesis, we measured *proportion of correct anticipatory looks* averaged across the two test trials. The proportional data were arcsine transformed and analysed using an independent samples *t*-test.[9] If in the test phase at least 50% of the infants in each group contributed anticipatory looks, and if the bilinguals made more correct anticipatory looks than the monolinguals, we may conclude that bilinguals respond more appropriately to more fragmented—or *less detailed*—visual stimuli.

### 2.2.3. Experiment 3: the gap-overlap task

The gap-overlap task (adapted from [23–26]) measures the ability to disengage attention from one visual stimulus and shift it towards a different visual stimulus.

#### 2.2.3.1. Stimuli
Three stimuli types were used: central fixation, peripheral target, and reward. The central fixation stimulus was a colourful 3 cm wide ($2.6° \times 2.6°$) animated cartoon of a clock. The peripheral target was a $2.6° \times 2.6°$ animated cartoon of a cloud. The reward was one of six $2.6° \times 2.6°$ animated cartoons (e.g. a balloon, car, butterfly). All visual stimuli flickered and were accompanied by a non-verbal sound (beep! or yip!) to attract the infant's attention.

#### 2.2.3.2. Procedure
Participants were presented with three trial conditions: baseline, gap and overlap. Each trial began with a centrally presented cartoon (the *central fixation stimulus* or *central stimulus*) that expanded and contracted for 0.8 s to attract the infant's attention. In the *baseline* and *gap* trials, once the child fixated on the central stimulus, the central stimulus vanished after 0.6–0.7 s. On its disappearance, a peripheral target was immediately presented in the baseline trials and after a 0.2 s delay in the gap trials. In the *overlap* trials, the central stimulus did not disappear; instead it ceased flickering, but remained onscreen and overlapped with the appearance of the target. It ceased flickering so the dynamic peripheral target was more interesting to the infant. The target was presented to either the left or the right of the central fixation stimulus at an eccentricity of 14.9° (17 cm). It remained onscreen until either the child looked at it, or until 3 s had elapsed. If the child looked at it within 1.2 s, she/he was rewarded by one of six animated cartoons (which appeared in place of the target). The time it took for the participant to shift his or her gaze to the peripheral target from the onset of the peripheral target was measured for each trial. Trials were presented in blocks of 12 until 12 'valid' trials per condition were acquired (see §2.2.3.3) or a maximum of 60 trials were presented. The whole procedure lasted less than 3 min.

---

[8]To detect a medium-sized effect ($d = 0.50$), we required a sample size of at least 27 per group (G*Power). Our sample size of 51 per group was sufficient to detect a small-medium effect ($d = 0.35$; when power $= 0.80$ and $\alpha = 0.05$; G*Power).

[9]Our sample size of 51 per group was sufficient to detect a medium-sized effect ($d = 0.50$; when power $= 0.8$ and $\alpha = 0.05$; G*Power).

### 2.2.3.3. Coding and analysis

Trials were automatically coded 'valid' if the infant fixated on the target after 0.2 s and before 1.2 s of its appearance [26,27]. If the participant failed to shift their gaze away from the central fixation stimulus within this time window, then the trial was automatically recorded as a 'failure to disengage'. In addition, trials were considered invalid if the participant failed to look at the central stimulus prior to the presentation of the target or if the child blinked or did not gaze towards the target.

To test our prediction that bilinguals are more likely to abandon the visual processing of a stimulus and thus shift attention faster to a novel stimulus than monolinguals, a 'disengagement' score was calculated by subtracting response times (RTs) in the baseline condition from RTs in the overlap condition, and compared across groups using an independent samples $t$-test.[9] To rule out the possibility that bilinguals merely saccade faster than monolinguals, an independent samples $t$-test was used to check that there was no significant group difference in RTs in the gap condition.[9]

## 2.2.4. Experiment 4: the visual memory task

The visual memory task probes whether bilingual infants shift attention more frequently and are less sensitive to the minute details of a visual stimulus than monolingual infants.

### 2.2.4.1. Stimuli

The stimuli included 15 blue line drawings (7.9 cm wide (7.0°)), four visual-auditory attention grabbers (the same as the ones in experiment 1), and gentle background music.

### 2.2.4.2. Procedure

The experiment consisted of 15 trials. All trials began with one of four attention grabbers centrally positioned on a white screen. After 1 s, the attention grabber was replaced with two blue line drawings, one on either side of the screen (left, right) and 13.2 cm (500 pixels, 11.6°) from the centre. The drawings remained onscreen for 5 s, after which the trial ended. In the first trial, the two drawings were identical; they were both line drawings of a man's head. In every subsequent trial, the drawing on one of the sides (e.g. the left) was replaced with a slightly different drawing. Over the course of the 15 trials, the drawing on one side of the screen remained the same, but the drawing on the other side of the screen gradually changed (in 14 steps) from a man's head to a woman holding flowers (see figure 3 for examples). Half the bilingual/monolingual infants saw the changes occur on the left side of the screen, half saw the changes on the right. Because experiment 4 did not contain any rewards, gentle background music was played throughout the experiment—to ensure that there was no silence and to create a 'warmer' environment. The procedure lasted about 75 s.

### 2.2.4.3. Coding and analysis

Two AOI were defined around the line drawings. The predefined AOIs extended 1 cm around the line drawings in case the infant focused on the outer edges of drawings and to account for small deviations in the quality of the calibration. Two measures were obtained: (i) number of times the participant switched visual attention between the two AOIs (divided by the time in seconds the participant spent looking at both AOIs), and (ii) proportion of time spent looking at the AOI that the novel stimuli appeared in (e.g. if for participant 1 it was the stimulus on the left side that kept changing over the course of the experiment, then for participant 1 we obtained proportion of time spent looking at the left AOI [left AOI/(left AOI + right AOI)]). For each measure, a mixed-effects regression model was carried out.[10] The predictor variables were *trial* (1–15) and *group* (monolingual versus bilingual). Random coefficients and an autoregressive covariance structure were assumed.

Although we expected both groups to switch between stimuli, we hypothesized that bilinguals would switch more frequently than monolinguals. The first regression model would test this hypothesis. Also, we argued that to build up detailed representations, infants must focus on one stimulus of interest at a time. If bilingual infants switch more frequently between stimuli, then they will spend less time focusing on the internal features of a single stimulus, and thus less time building up a detailed model of any one stimulus. Therefore, we hypothesized that monolinguals would focus more on the internal features of a

---

[10]Simulation studies suggest that 15 observations per individual (1530 observations in total) should provide sufficient power (greater than 0.80 [22]).

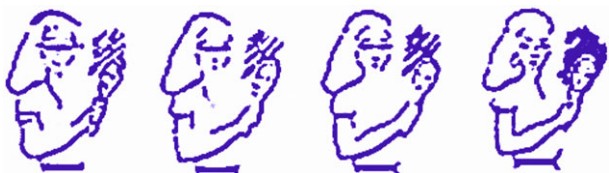

**Figure 3.** Experiment 4 consisted of 15 trials. In each trial, two stimuli were presented, one on either side of the screen. In the first trial, the two stimuli were identical (the leftmost drawing). Over the course of the 15 trials, the stimulus on one side of the screen remained the same, but the stimulus on the other side of the screen changed. For example, in trial 5 the drawing second from the left was presented with the leftmost drawing; in trial 10, the drawing third from the left was presented with the leftmost drawing; and in trial 15, the drawing on the far right was presented with the leftmost drawing.

stimulus (rather than spending time switching between the two stimuli) and thus notice the change sooner (albeit with fewer switches between stimuli) than bilinguals. The second regression model would test this hypothesis by ascertaining whether the infants looked at one of the stimuli for significantly longer than the other (indicating that they discriminated between the two) and whether this happened sooner for monolinguals than bilinguals. If we found a statistically significant effect of group, then we followed it up with one-sample $t$-tests to find out whether time spent looking at the novel stimulus was greater than expected by chance (0.50).[11] That is, we would ascertain whether the monolingual infants were better than the bilingual infants at remembering that the stimulus on one side was the same as in previous trials, while the stimulus on the other side had changed.

Moreover, we hypothesized that (even after noticing a change) over the duration of the experimental task, the monolinguals would spend significantly more time than the bilinguals processing the familiar (versus novel) stimulus. In other words, monolinguals would err on the side of consolidating their knowledge, while bilinguals would err on the side of exploration. The second regression model would also test this hypothesis.

### 2.2.5. Language and socioeconomic status background information

*Proportion of English language input* was calculated using the LEQ [28]. The purpose of the LEQ was to calculate the amount of exposure to English the child was hearing in a typical week. It was administered in the form of a 10 min interview through discussion with the parent. Information from the parent was entered into an online assessment tool (www.psy.plymouth.ac.uk/LEQ/) which automatically yielded an 'exposure to English' percentage score. We used this information to assess exclusion language criteria.

Parents also completed the Oxford CDI (OCDI; [29]) for each language their child was exposed to. The OCDI is a list of 416 words that are commonly acquired in infancy. For each word, the carer indicated whether their child can 'understand' and/or 'understand and say' the word. We used the OCDI to confirm that their infant was not yet producing words. If there were any group differences, then we could explore the effect in bilinguals using the receptive language measure (number of words understood). We would expect that the more words a bilingual child understands from two or more languages, the more they are 'embedded' in bilingualism. It would be interesting to use it as a continuous measure of bilingualism and find out whether it covaries with the observed effect. We would do this in a separate 'exploratory' section. The OCDI is available to view online and download from the Oxford Babylab website (https://www.psy.ox.ac.uk/research/oxford-babylab/research-overview/oxford-cdi).

Infants' background information, including their parents' SES, was gathered using a parent-report questionnaire (electronic supplementary material). We used the information to match participants on SES.

### 2.3. Statistical analyses

Because we were only interested in group differences, for all experimental trials, individual data points that lie greater than ±3 s.d. from the group mean were excluded from analyses. Data were analysed and visualized using R [30].

---

[11]To detect a medium-sized effect ($d = 0.50$), we required a sample size of at least 27 per group (G*Power). Our sample size of 51 per group was sufficient to detect a small-medium effect ($d = 0.35$; when power = 0.8 and $\alpha = 0.05$; G*Power).

**Table 2.** The relationship between pre-switch trials and correct anticipatory looks. (The 'interaction' model fit the data better than the '+ trial' model, $\chi^2_2 = 6.43$, $p = 0.040^*$. $^*p < 0.05$, $^{***}p < 0.001$.)

| model | d.f. | AIC | BIC | logLik | deviance | $\chi^2$ | $\chi^2$d.f. | $p$ |
|---|---|---|---|---|---|---|---|---|
| null | 2 | 1195.5 | 1205.2 | −595.76 | 1191.5 | | | |
| + trial | 3 | 1174.1 | 1188.6 | −584.06 | 1168.1 | 23.40 | 1 | <0.001*** |
| + group | 4 | 1175.0 | 1194.3 | −583.49 | 1167.0 | 1.15 | 1 | 0.284 |
| interaction | 5 | 1171.7 | 1195.8 | −580.85 | 1161.7 | 5.28 | 1 | 0.022* |

**Table 3.** Estimated fixed effects (model: + trial).

| effect | estimate | s.e | Z |
|---|---|---|---|
| intercept | −1.12 | 0.19 | −5.79 |
| trial | 0.14 | 0.03 | 4.78 |

# 3. Results

## 3.1. Experiment 1

To check that the infants could anticipate the appearance of a reward, for the pre-switch trials we fitted a mixed effects logistic model (the 'null' model) with 'correct anticipatory look' as the outcome variable and random intercepts for participants. We assumed random intercepts for participants because it is likely that baseline 'anticipatory looking' varies across infants irrespective of group. We then added a fixed effect of trial (the '+ trial' model):

$$\text{model (null): correct anticipatory look} \sim (1|\text{participant}) + \varepsilon$$
$$\text{model (+ trial): correct anticipatory look} \sim (1|\text{participant}) + \text{trial} + \varepsilon.$$

If infants learned to anticipate the appearance of the reward, then the addition of 'trial' should significantly improve the 'null' model. Comparisons of Akaike's information criterion (AIC) and Schwarz's Bayesian information criterion (BIC), as well as a likelihood ratio test, show a highly significant effect of trial (table 2). This suggests that the task was working (see table 3 for the estimated fixed effect of trial).

Adding a fixed effect of group did not improve the model (table 2; see also figure 4a). However, a comparison of AIC and a likelihood ratio test suggest that adding the possibility of an interaction between group and trial (the 'interaction' model) does improve the '+ group' model ($p = 0.022$). Moreover, the 'interaction' model fits the data better than the (no-group) '+ trial' model (table 2). This hints at the possibility that bilinguals may learn to anticipate rewards at a faster rate than monolinguals. Interestingly, bilingual infants ($M = 0.28$, s.d. $= 0.30$) were less likely than monolingual infants ($M = 0.42$, s.d. $= 0.35$) to make correct anticipatory looks during the first three trials, $t_{100} = 2.14$, $p = 0.035$, but caught up with the monolinguals by the final three trials ($0.52$, s.d. $= 0.36$ (bilinguals) versus $0.48$, s.d. $= 0.33$ (monolinguals)), $t_{100} = 0.67$, $p = 0.504$.

It is important to note that a comparison of BIC does not favour the 'interaction' model over the '+ trial' model. This may be because BIC penalizes additional parameters more heavily than AIC. On the one hand, the '+ trial' model *is* more parsimonious than the 'interaction' model. On the other hand, the reality may be complex and require additional parameters. In any case, we advise caution when interpreting these results.

To replicate the finding by Kovacs & Mehler [9] that bilinguals but not monolinguals learn to inhibit a learned behaviour, we fitted a linear mixed effects model (the 'null' model) to the post-switch trials data, with 'correct anticipatory look' as the outcome variable and random intercepts for participants. We then added a fixed effect of trial (the '+ trial' model). Comparisons of AIC and BIC, as well as a likelihood ratio test, show a highly significant effect of trial (table 4). However, the addition of a fixed effect of 'group' did not greatly improve the model (table 4). Finally, we predicted an interaction between group and trials,

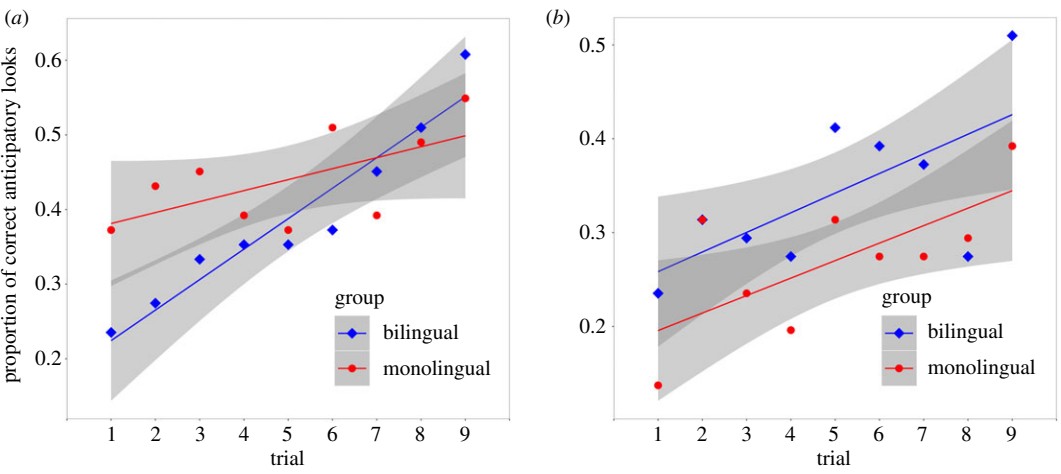

**Figure 4.** (*a*) In trial 1 of the pre-switch block, 24% of bilingual infants and 37% of monolingual infants made correct anticipatory looks. By trial 9, 61% of bilingual infants and 55% of monolingual infants made correct anticipatory looks. The increase in anticipatory looks across trials was statistically significant, indicating learning. (*b*) In trial 1 of the post-switch block, only 24% of bilingual infants and 14% of monolingual infants made correct anticipatory looks. By trial 9, this had increased to 51% in bilinguals and 39% in monolinguals, indicating that both groups were successfully redirecting their anticipatory looks (experiment 1).

**Table 4.** The relationship between post-switch trials and correct anticipatory looks. ($^{\#}p < 0.10$, $^{***}p < 0.001$.)

| model | d.f. | AIC | BIC | logLik | deviance | $\chi^2$ | $\chi^2$d.f. | $p$ |
|---|---|---|---|---|---|---|---|---|
| null | 2 | 1087.5 | 1097.1 | −541.75 | 1083.5 | | | |
| + trial | 3 | 1076.2 | 1090.7 | −535.09 | 1070.2 | 13.31 | 1 | <0.001*** |
| + group | 4 | 1075.3 | 1094.6 | −533.66 | 1067.3 | 2.87 | 1 | 0.090# |
| interaction | 5 | 1077.3 | 1101.4 | −533.66 | 1067.3 | <0.01 | 1 | 0.988 |

with bilinguals (but not monolinguals) learning to inhibit their learned behaviour and make more post-switch correct anticipatory looks. However, the addition of an interaction did not improve the model (table 4). In short, our data do not support the claim that bilingual infants but not monolingual infants can inhibit learned behaviour; proportion of correct anticipatory looks increased in both groups.

## 3.2. Exploratory analyses

To help interpret the data, we ran similar analyses as before, but this time, we analysed only looks captured within the anticipatory period (see §2.2.1.2). This is a 'purer' measure of anticipatory looking because the target was not onscreen at any point during the anticipatory period. Also, rather than analyse 'correct anticipatory look', we measured the difference (in seconds) between correct and incorrect anticipatory looking durations. We thus fitted a linear mixed effects model with 'difference in seconds' as the outcome variable and random intercepts for participants—and then, as before, added a fixed effect of trial, a fixed effect of group and finally a trial-by-group interaction. For both pre-switch trials and post-switch trials (tables 5 and 6), the results were similar to those in §3.1.

## 3.3. Experiment 2

To check that the infants learned to associate the cue (elephant, snowman) with the location of a reward, we fitted a mixed effects logistic model (the 'null' model) with 'correct anticipatory look' as the outcome variable and random intercepts for participants. We then added a fixed effect of trial (the '+ trial' model):

$$\text{model (null): correct anticipatory look} \sim (1|\text{participant}) + \varepsilon$$
$$\text{model (+ trial): correct anticipatory look} \sim (1|\text{participant}) + \text{trial} + \varepsilon.$$

**Table 5.** The relationship between pre-switch trials and correct anticipatory looks. (The 'interaction' model fit the data better than the '+ trial' model, $\chi^2_2 = 9.94$, $p = 0.007$**. ** $p < 0.01$, *** $p < 0.001$.)

| model | d.f. | AIC | BIC | logLik | deviance | $\chi^2$ | $\chi^2$d.f. | $p$ |
|---|---|---|---|---|---|---|---|---|
| null | 3 | −2479.2 | −2464.8 | 1242.6 | −2485.2 | | | |
| + trial | 4 | −2492.5 | −2473.2 | 1250.2 | −2500.5 | 15.25 | 1 | <0.001*** |
| + group | 5 | −2491.6 | −2467.5 | 1250.8 | −2501.6 | 1.11 | 1 | 0.292 |
| interaction | 6 | −2498.4 | −2469.5 | 1255.2 | −2510.4 | 8.84 | 1 | 0.003** |

**Table 6.** The relationship between post-switch trials and correct anticipatory looks. ($^{\#}p < 0.10$, *** $p < 0.001$.)

| model | d.f. | AIC | BIC | logLik | deviance | $\chi^2$ | $\chi^2$d.f. | $p$ |
|---|---|---|---|---|---|---|---|---|
| null | 3 | −2486.5 | −2472.0 | 1246.2 | −2492.5 | | | |
| + trial | 4 | −2502.6 | −2483.3 | 1255.3 | −2510.6 | 18.16 | 1 | <0.001*** |
| + group | 5 | −2503.5 | −2479.4 | 1256.8 | −2513.5 | 2.88 | 1 | 0.090$^{\#}$ |
| interaction | 6 | −2501.9 | −2473.0 | 1257.0 | −2513.9 | 0.39 | 1 | 0.534 |

**Table 7.** The relationship between trials and correct anticipatory looks.

| model | d.f. | AIC | BIC | logLik | deviance | $\chi^2$ | $\chi^2$d.f. | $p$ |
|---|---|---|---|---|---|---|---|---|
| null | 2 | 1719.2 | 1729.5 | −857.58 | 1715.2 | | | |
| + trial | 3 | 1721.0 | 1736.5 | −857.50 | 1715.0 | 0.16 | 1 | 0.685 |
| +group | 4 | 1723.0 | 1743.6 | −857.48 | 1715.0 | 0.03 | 1 | 0.873 |

If infants learned to anticipate the appearance of the reward, then the addition of 'trial' should significantly improve the 'null' model. However, comparisons of AIC and BIC, as well as a likelihood ratio test, show that this was not the case (table 7). This suggests that the task was too difficult for the infants. (For sake of completeness, we added a fixed effect of group, but this did not improve the model; table 7.)

## 3.4. Experiment 3

To test our prediction that bilinguals are more likely to abandon the visual processing of a stimulus and thus shift attention faster to a novel stimulus than monolinguals, a 'disengagement' score was calculated by subtracting RTs in the baseline condition from RTs in the overlap condition. Each participant provided at least six valid trials per condition (baseline, overlap). The bilingual infants did not disengage significantly faster than the monolingual infants, $t_{99} = 1.32$, $p = 0.190$.

Because the data were significantly non-normal—especially the monolingual data ($Z_{Skewness} = 2.79$, $Z_{Kurtosis} = 1.91$, $D_{100} = 0.15$, $p = 0.005$), we decided to also carry out a non-parametric test, the Mann–Whitney U-test. This is because we could not satisfactorily logarithmically transform the data as some of the data contained negative values. The non-parametric test confirmed that the bilingual infants did not disengage significantly faster than the monolingual infants, $U = 1563$, $z = 1.76$, $p = 0.079$ (figure 5)—though it may be important to note that the one-tailed significance value was $p = 0.040$.

We then checked that the two groups did not significantly differ on gap RT. As expected, they did not, $t_{97} = 0.62$, $p = 0.540$. This was confirmed with a non-parametric test, $U = 1365$, $z = 0.43$, $p = 0.666$. It suggests that the ocular-motor system is similar in both groups. (Note: two of the monolingual infants and one of the bilingual infants who provided at least six valid trials in the baseline and overlap conditions provided only five valid trials in the gap condition; however, when we replaced these participants with three participants who provided at least six valid trials in the gap condition, the result was the same: $t_{96} = 0.11$, $p = 0.911$; $U = 1388$, $z = 0.59$, $p = 0.558$.)

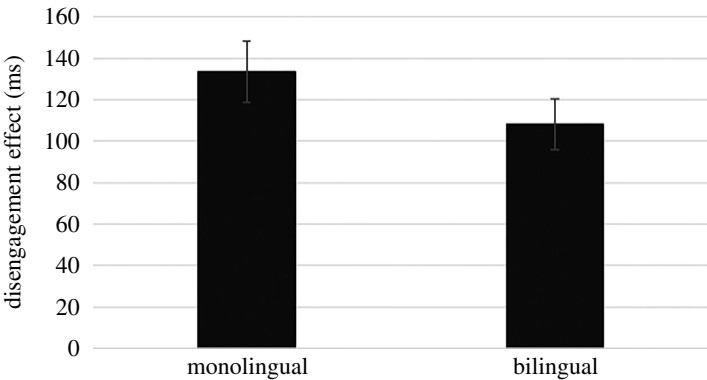

**Figure 5.** The bilingual infants ($n = 51$) were not significantly faster than the monolingual infants ($n = 51$) ($p = 0.079$, two-tailed). Error bars represent $\pm 1$ standard error of the mean (experiment 3).

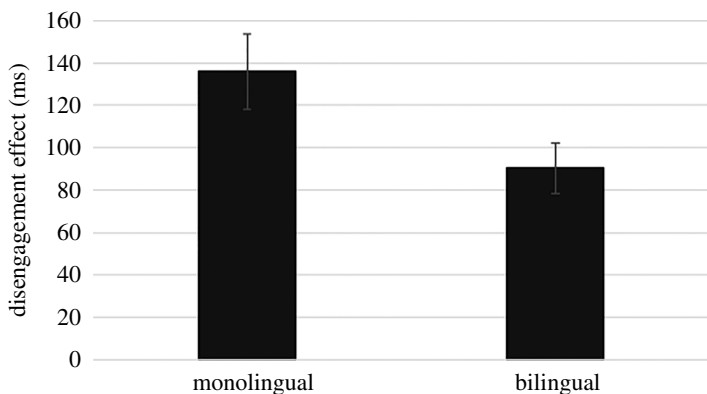

**Figure 6.** Analysis of the best quality data (at least 12 valid trails in each condition) suggests that bilingual infants ($n = 31$) disengage attention significantly faster than monolingual infants ($n = 34$). Error bars represent $\pm 1$ s.e. of the mean (experiment 3).

### 3.4.1. Exploratory analyses

During testing, we realized that the data we were collecting were of lower quality than expected. For example, trials were presented in blocks of 12 until 12 'valid' trials per condition were acquired or a maximum of 60 trials were presented (see §2.2.3.2). We expected most infants to provide 12 valid trials per condition by the fifth and final block of trials, but in a large number of cases ($n = 49$) they did not. We still obtained at least six valid trials per condition (baseline, overlap) from these children, but we expected more. We felt that we should reanalyse the data, but this time include only infants who provided 12 valid trials per condition (31 bilinguals, 34 monolinguals). As expected, the groups did not significantly differ on gap RT, $t_{63} = 0.26$, $p = 0.794$ ($U = 544.00$, $z = 0.22$, $p = 0.823$), indicating similar ocular-motor systems. But the bilingual infants did disengage attention significantly faster than the monolingual infants, $t_{63} = 2.10$, $p = 0.040$; $U = 715.00$, $z = 2.47$, $p = 0.014$ (figure 6).

Because the result of our exploratory analysis was statistically significant, we decided to look more closely at the relationship between language exposure and the ability to disengage. Any child who was even slightly exposed to a second language was included in a linear regression, with language exposure as the predictor and the disengagement effect as the outcome variable. The data (all positive values) were logarithmically transformed (base 10). We found that the less exposure to English that a child had (as measured using the LEQ), the faster the infant was at disengaging attention to shift attention to a peripheral visual stimulus, $F_{1,34} = 6.35$, $p = 0.017$ (table 8 and figure 7; non-transformed data: $B = 0.93$, s.e. $B = 0.39$, $\beta = 0.38$; $F_{1,34} = 5.72$, $p = 0.022$). Although this analysis is exploratory, it is not something that has been reported in the literature and is worth investigating in a future study.

We ran the same regression with the untransformed data from the original (larger) sample of participants and obtained a similar result: $F_{1,55} = 8.42$, $p = 0.005$ (table 9; see also figure 8). The data

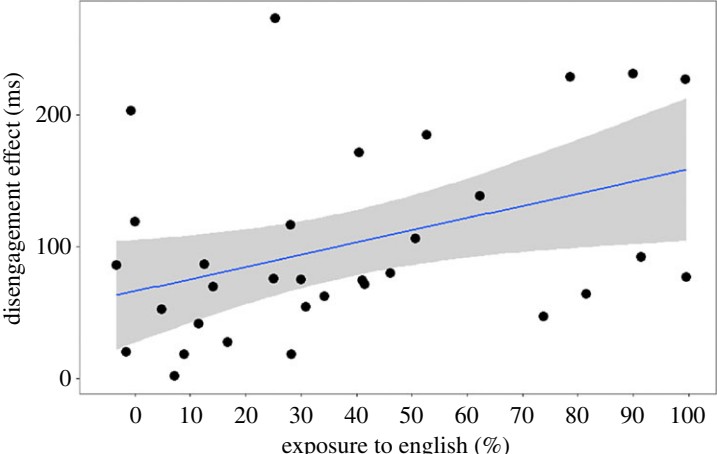

**Figure 7.** The more exposure to English the infant had (as estimated using the LEQ), the slower the infant was at disengaging attention in order to shift attention to a peripheral stimulus ($n = 36$) (experiment 3). Any child who was even slightly exposed to a second language was included in this regression.

**Table 8.** Regression analysis: the disengagement effect as a function of bilingualism. (* $p < 0.05$.)

|  | B | s.e. B | β |
|---|---|---|---|
| intercept | 1.67 | 0.10 |  |
| bilingualism | 0.01 | <0.01 | 0.40* |

included a large negative value, unsuitable for a logarithmic transformation, but non-parametric correlations confirm the relationship: $r_s = 0.36$, $p = 0.006$; $\tau = 0.26$, $p = 0.004$.

## 3.5. Experiment 4

The visual memory task probed whether bilingual infants (i) shift attention more frequently, and (ii) are less sensitive to the minute details of a visual stimulus than monolingual infants.

### 3.5.1. Do bilingual infants switch attention more frequently?

To probe whether bilingual infants shift attention more frequently than monolingual infants, we fitted a linear mixed effects model with number of switches as the outcome variable, a fixed effect of trial, a fixed effect of group, and random intercepts for participants. We assumed random intercepts for participants because it is likely that baseline 'switching' varies across infants irrespective of group. To isolate any effect of 'group', we compared this (group) model with a reduced no-group (null) model:

$$\text{model (group): no. of switches} \sim \text{trials} + (1|\text{participant}) + \text{group} + \varepsilon$$
$$\text{model (null): no. of switches} \sim \text{trials} + (1|\text{participant}) + \varepsilon.$$

A comparison of AIC and BIC, as well as a likelihood ratio test (table 10), shows that the bilingual infants shifted attention more often during the task than the monolingual infants (see table 11 for the estimated fixed effect of bilingualism; see also figure 9).

However, it is possible that the bilingual infants merely spent more time looking at the stimuli—and thus had more time to shift attention—than the monolingual infants. It was therefore necessary to analyse 'number of switches' relative to 'total time spent looking at both stimuli'. We fitted a linear mixed effects model with 'number of switches divided by total time spent looking at both stimuli' as the outcome variable, a fixed effect of trial, and random intercepts for participants (model 0). We then added 'group' as an additional fixed effect (model 1):

$$\text{model 0 (null): (no. of switches/total looking time)} \sim \text{trial} + (1|\text{participant}) + \varepsilon$$
$$\text{model 1 (group): (no. of switches/total looking time)} \sim \text{trial} + \text{group} + (1|\text{participant}) + \varepsilon.$$

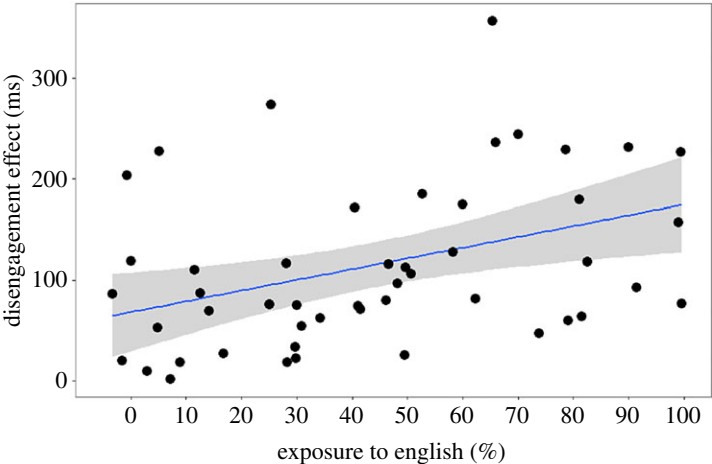

**Figure 8.** Analysis of the larger dataset ($n = 57$) supports the exploratory finding that the infant's exposure to English (as estimated using the LEQ) is correlated with their ability to disengage attention (experiment 3). Any child who was even slightly exposed to a second language was included in this regression.

**Table 9.** Regression analysis: the disengagement effect as a function of bilingualism. ($**p < 0.01$.)

|  | B | s.e. B | β |
|---|---|---|---|
| intercept | 68.40 | 19.45 |  |
| bilingualism | 1.07 | 0.37 | 0.37** |

**Table 10.** The relationship between group (bilingual and monolingual) and number of switches. ($***p < 0.001$.)

| model | d.f. | AIC | BIC | logLik | deviance | $\chi^2$ | $\chi^2$d.f. | p |
|---|---|---|---|---|---|---|---|---|
| null | 4 | 4299.4 | 4320.3 | −2145.7 | 4291.4 |  |  |  |
| + group | 5 | 4272.9 | 4299.1 | −2131.5 | 4262.9 | 28.45 | 1 | <0.0001*** |

**Table 11.** Estimated fixed effects (model: group).

| effect | estimate | s.e. | t |
|---|---|---|---|
| intercept | 2.00 | 0.09 | 21.56 |
| bilingualism | 0.61 | 0.11 | 5.74 |
| trial | −0.09 | 0.01 | −13.32 |

We compared AIC and BIC, and carried out a likelihood ratio test, between the two models. AIC (but not BIC) favoured the model with 'group'. The likelihood ratio test was statistically significant (table 12). Adding 'group' improved the 'null' model (see table 13 for the estimated fixed effect of bilingualism). These data support our hypothesis that bilingual infants switch more than monolingual infants (figure 10). Because the change in stimulus across trials was not strictly linear, we also tested for any inter-dependence between trials and group. However, neither comparisons of AIC/BIC nor the likelihood ratio test favoured the interaction model (table 12).

### 3.5.2. Do bilinguals spend less time processing familiar stimuli? Are they worse at remembering details?

To check that the infants noticed that the stimulus on one side of the screen was gradually changing, we fitted a linear mixed effects model (the 'null' model) with 'proportion of looking to novel stimuli' as the outcome

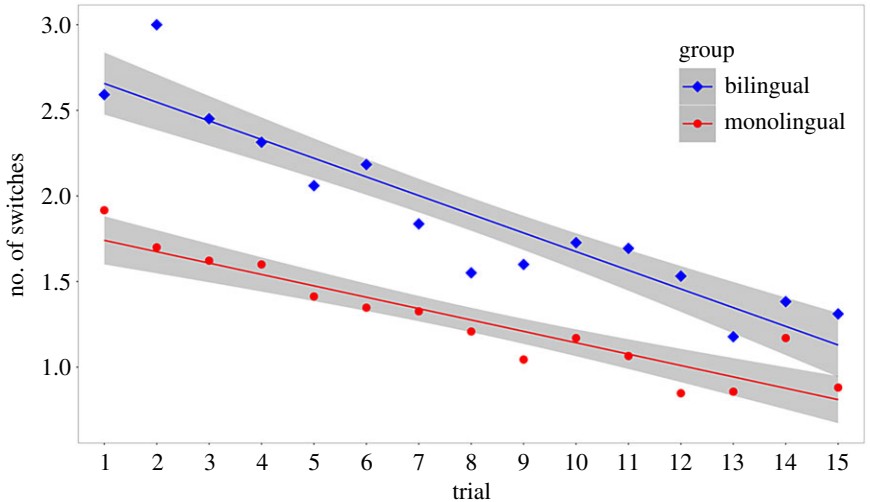

**Figure 9.** The relationship between trial and number of switches, by group (bilingual and monolingual; experiment 4).

**Table 12.** The relationship between group (bilingual and monolingual) and proportion of switches. ($**p < 0.01$.)

| model | d.f. | AIC | BIC | logLik | deviance | $\chi^2$ | $\chi^2$d.f. | $p$ |
|---|---|---|---|---|---|---|---|---|
| null | 4 | 3088.8 | 3109.6 | −1540.4 | 3080.8 | | | |
| + group | 5 | 3083.7 | 3109.7 | −1536.8 | 3073.7 | 7.12 | 1 | 0.008** |
| interaction | 6 | 3085.3 | 3116.6 | −1536.7 | 3073.3 | 0.35 | 1 | 0.552 |

**Table 13.** Estimated fixed effects (model 1).

| effect | estimate | s.e. | $t$ |
|---|---|---|---|
| intercept | 1.29 | 0.07 | 19.36 |
| monolingual | −0.22 | 0.08 | −2.71 |
| trial | −0.03 | 0.005 | −6.46 |

variable and random intercepts for participants. We then added a fixed effect of trial (the '+ trial' model):

model (null): proportion of looking to novel stimuli $\sim$ (1|participant) + $\varepsilon$

model (+ trial): proportion of looking to novel stimuli $\sim$ (1|participant) + trial + $\varepsilon$.

If infants learned that the stimulus on one side of the screen was gradually changing, then the addition of 'trial' should significantly improve the 'null' model. However, comparisons of AIC and BIC, as well as a likelihood ratio test, show that this was not the case (table 14). This suggests that the task was too difficult for the infants. For sake of completeness, we added a fixed effect of group. Although the monolingual infants spent more time than the bilingual infants looking at the familiar stimulus (relative to the novel stimuli), which was what we predicted, the addition of 'group' did not improve the model, so we cannot draw firm conclusions from this analysis (table 14).

### 3.5.3. Exploratory analyses

Our results suggest that infants from bilingual homes switch attention more frequently than infants from monolingual homes. Experiment 3 suggests that they may also be quicker at disengaging attention from visual stimuli in order to shift attention to new visual stimuli. This raises the question of whether the ability to disengage attention is related to number of switches in bilingual infants?

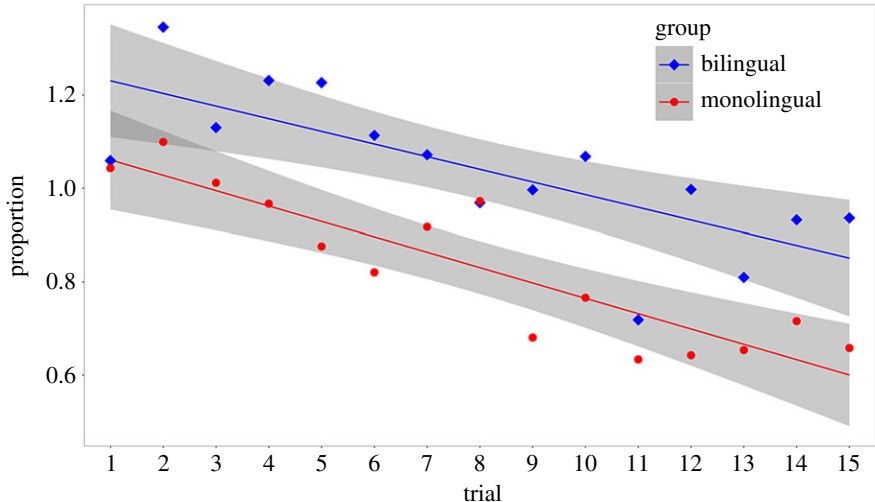

**Figure 10.** The relationship between trial and proportion of switches to time spent looking at both stimuli, by group (bilingual and monolingual; experiment 4).

**Table 14.** The relationship between trials and proportion of looking to novel stimuli.

| model | d.f. | AIC | BIC | logLik | deviance | $\chi^2$ | $\chi^2$d.f. | $p$ |
|---|---|---|---|---|---|---|---|---|
| null | 3 | 621.14 | 636.99 | −307.57 | 615.14 | | | |
| + trial | 4 | 622.80 | 643.93 | −307.40 | 614.80 | 0.34 | 1 | 0.559 |
| + group | 5 | 624.39 | 650.80 | −307.19 | 614.39 | 0.41 | 1 | 0.523 |

To test the relationship between attentional disengagement and switching behaviour, we analysed the data from experiments 3 and 4. Thirty-two bilingual infants and 37 monolingual infants provided good data for both experiments and were thus included in this exploratory analysis. Data from one bilingual infant and three monolingual infants were removed for having data points greater than ±2 s.d. from the mean. Because the disengagement RT data were non-normal in the bilingual group, $D_{31} = 0.20$, $p = 0.003$, non-parametric tests were carried out. We report both Kendall's and Spearman's correlation coefficients because while the former is possibly better for small samples [31], the latter is easier to interpret. Disengagement RTs were not related to number of switches in the combined group of 65 infants, $\tau = -0.16$, $p = 0.066$ ($r_s = -0.23$, $p = 0.064$). However, disengagement RTs were negatively correlated with number of switches in the bilingual infants, $\tau = -0.30$, $p = 0.019$ ($r_s = -0.41$, $p = 0.022$), but not in the monolingual infants, $\tau = 0.03$, $p = 0.824$ ($r_s = 0.02$, $p = 0.912$) (figure 11).

A similar pattern was observed when the relationship between disengagement RTs and 'proportion of switches to face looking' was analysed. Data from two bilingual infants and five monolingual infants were removed for having data points greater than ±2 s.d. from the mean. In the single combined group of 62 infants, there was no relationship, $\tau = -0.03$, $p = 0.775$ ($r_s = -0.02$, $p = 0.862$). However, there was a relationship in bilingual infants, $\tau = -0.32$, $p = 0.013$ ($r_s = -0.41$, $p = 0.025$), but not in monolingual infants, $\tau = 0.20$, $p = 0.105$ ($r_s = 0.28$, $p = 0.125$) (figure 12).

These exploratory data hint that the ability to disengage attention is related to switching behaviour in infants from bilingual homes but not necessarily in infants from monolingual homes.

# 4. Conclusion

Our data do not support the claim that infants raised in bilingual homes are better at inhibiting a learned behaviour than infants raised in monolingual homes (experiment 1). However, our data suggest that infants raised in bilingual homes are faster at disengaging attention in order to shift attention to a new stimulus (experiment 3) and switch attention more frequently between two visual stimuli (experiment 4). Furthermore, exploratory analyses suggest a relationship between speed of visual

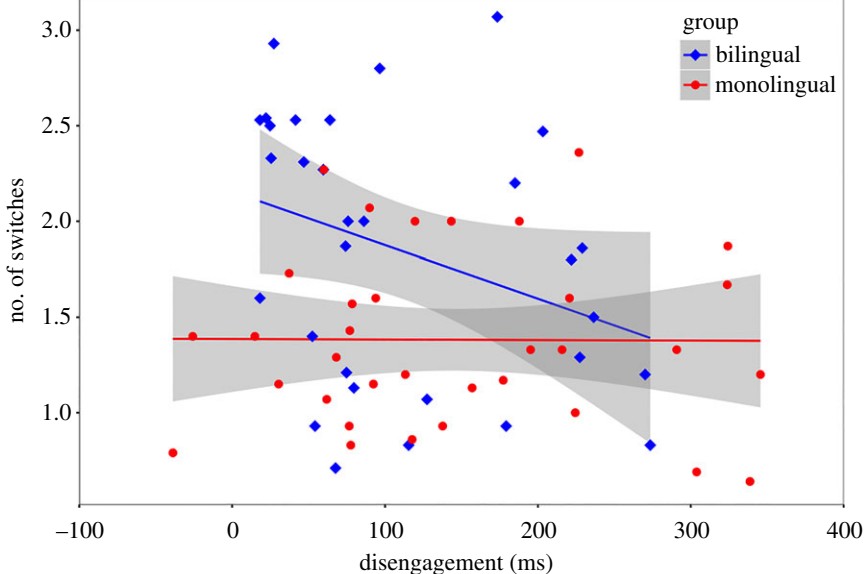

**Figure 11.** The relationship between the ability to disengage attention (in ms) and number of attentional switches, by group (bilingual and monolingual). A relationship was observed in bilingual ($r_s = -0.41$), but not monolingual ($r_s = 0.02$), infants (experiments 3 and 4).

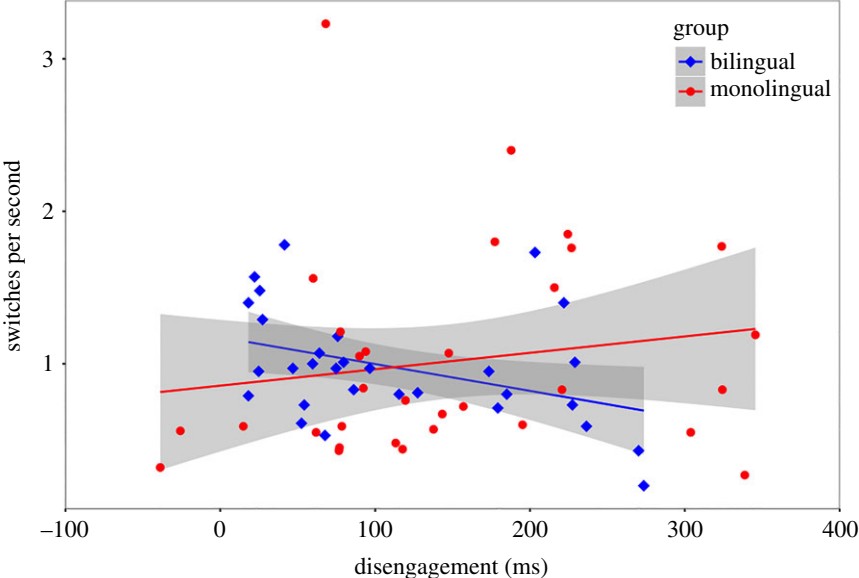

**Figure 12.** The relationship between the ability to disengage attention (in ms) and number of attentional switches s$^{-1}$ of looking time, by group (bilingual and monolingual). A relationship was observed in bilingual ($r_s = -0.41$), but not monolingual ($r_s = 0.28$), infants (experiments 3 and 4).

disengagement (experiment 3) and number of switches between two visual stimuli (experiment 4). This raises the possibility that infants adapt to bilingual environments partly by disengaging attention faster and switching attention more frequently. It supports our proposal that bilingual infants adapt by placing more weight on novel information in order to collect more samples from their more varied environments.

Unfortunately, we were unable to ascertain whether bilingual infants respond more appropriately to more fragmented—or less detailed—visual stimuli. Nor were we able to ascertain whether they spent less time visually processing a familiar stimulus or whether they were worse at remembering the details of a visual stimulus. So, we cannot make any claims about how they modelled or represented information. Our hypothesis that bilingual infants get by on less detailed models of the environment remains untested.

# 5. Discussion

Is mere exposure to bilingual environments enough? We suggest that it is. We found that infants exposed to bilingual environments switch attention more frequently than infants exposed to monolingual environments. Exploratory analyses also suggest that switching attention is related to the ability to disengage attention in order to shift attention to a new visual stimulus—in infants raised in bilingual, but not monolingual, homes. This tells us that infants adapt to their different language environments. Also, because the infants had not yet begun to speak, it tells us that mere exposure to a second language is sufficient to observe a difference. That is, the difference is a result of hearing not producing two or more languages.

However, the explanation for this difference is not clear. It is likely that infants who regularly hear two or more languages are exposed to a more variable and less predictable language (and possibly sociocultural) environment than infants who hear only one language. We therefore speculate that infants exposed to bilingual environments adapt by exploring (sampling) their environment more, by placing more weight on information-seeking. This would explain why they might disengage attention faster and switch attention more frequently than infants from monolingual homes.

How would these behavioural differences come about? Insight may come from two infant studies which demonstrated that selective attention to a talking mouth (versus eyes) facilitates concurrent language learning [32] and predicts later expressive language development [33]. These studies support Lewkowicz & Hansen-Tift's [34] claim that infants use redundant audiovisual speech cues to learn words by matching sounds to lip movements. Perhaps, because of their more complex language environments, bilingual infants actively seek out multiple sources of information. We therefore speculate that infants raised in more variable language environments collect more samples from their environment. Perhaps bilingual infants are more likely to disengage the focus of their attention from a stimulus (e.g. a toy) in order to shift it towards the mouths, facial expressions or bodily movements of the various speakers in their environment; and maybe they switch more frequently between these different sources of information. It would be interesting to explore this hypothesis by observing the infants in their homes.

Further insight may come from our suggestion that bilingual infants outperform monolingual infants on the gap-overlap task. In our study, saccadic reaction times did not significantly differ across groups in the gap condition, but they did in the overlap (minus baseline) condition. This difference suggests that visual orienting may be underpinned by differences in specific neuro-circuitry rather than a function of whole-brain efficiency. While the generation of saccades—reflected by saccadic reaction times in the gap condition—is mediated by subcortical processes and the development of corticospinal tracts connecting brain stem to cerebral cortex [35], the ability to disengage attention—reflected by saccadic reaction times in the overlap condition—is associated with development of the splenium, the thickest part of the corpus callosum [36]. Little is known about the development of the splenium in human infants, but in rhesus monkeys the splenium is the site of rapid axonal elimination during the first few months of life [37]. This occurs at a time when the rhesus macaque brain is particularly malleable to external stimuli which help shape large parts of its cerebral cortex [38]. It is therefore conceivable that changes in the human infant's language environment could adaptively drive exploratory behaviours and the early development of the splenium. Indeed, language production at 24 months has been associated with rate of change in splenium development (but not in other white matter tracts) from six months of age [39], and variation in word acquisition has been linked to variation in attentional orienting [40] and visual experience [41], so it is possible that differences in visual orienting and splenium development reflect interdependent adaptive processes through which an infant's internal neuro-circuitry calibrates to the metrics of the external world. This would explain why variation in early social interaction is associated with attentional disengagement later in development [42] and it fits with evidence that corpus collosum is better preserved in elderly lifelong bilinguals [43].

However, most studies in the literature explain differences between bilinguals and monolinguals as a difference in inhibitory control (see [1], for review). This comes from the idea that the purported bilingual advantage is the result of managing two or more languages during language production. Our data do not support this hypothesis because our participants were preverbal. But even if the inhibitory control theory were modified to include any advantage in cognitive control or cognitive flexibility as a result of greater complexity in the bilingual environment, our data would not support it for this age. This is because we could not replicate Kovacs & Mehler's [9] finding that only bilingual infants can inhibit a learned behaviour ($n = 20$). Although Kovacs & Mehler [9] found that post-switch anticipatory responses increased in the bilingual, not monolingual, group, they did not report whether the

bilingual infants made more anticipatory looks than expected by chance. Their results were, however, partly supported by Comishen et al. [44], who ran a similar study with six-month-old infants ($n = 20$). Comishen et al. [44] found that post-switch anticipatory looks were not significantly different from chance in monolinguals but were in bilinguals. However, unlike Kovacs and Mehler, they did not demonstrate any significant difference in anticipatory looks between the monolinguals and bilinguals. Their groups' error bars (which represent one standard error of the mean) overlap, which confirms that there was no statistically significant difference between the two groups. A more direct replication attempt was carried out by Tsui & Fennell [45], who tested older bilingual infants (nine months; $n = 23$) using the same task as Kovacs & Mehler [9] and us. In their study, bilingual infants neither looked at the correct location more nor learned the new pattern-reward association faster. They found no evidence that bilingual language processing strengthens inhibitory control mechanisms in preverbal infants. Why might Comishen et al. find a difference, but not Tsui and Fennell or us? There could be differences in methods across laboratories. For example, Comishen et al. [44] excluded from analysis the first 10 trials and the last 10 trials of their study, while Tsui and Fennell excluded no trials but presented only six trials per block. In our much larger replication, both bilingual *and* monolingual infants could inhibit a learned behaviour.

Although neither Comishen et al. [44] nor Tsui & Fennell [45] demonstrated post-switch differences in anticipatory looks *between groups*, and thus no direct evidence that bilingual language processing strengthens inhibitory processes in preverbal infants, Comishen et al. did report a post-switch group difference in mean 'reactive' latencies—eye movements occurring between 133 ms after target onset and 133 ms after target offset. Reactive latencies were faster in bilingual infants than monolingual infants. This may reflect expectations that could not quite manifest fast enough into anticipatory looks. Alternatively, it suggests that infants exposed to bilingual environments become quicker at disengaging visual attention from one stimulus in order to shift attention to a new one. This would fit our theory and data. We could not test our hypothesis that bilingual infants construct less detailed models of their environments (because the task was too difficult for them), but we were able to investigate whether infants from bilingual homes are faster at abandoning the visual processing of a stimulus to shift attention to a novel stimulus; and they seemed to. Therefore, our data broadly support our proposal that exposure to more varied language environments drive infants to explore (sample) further by placing more weight on new information and switching attention more. This dovetails with Singh et al.'s [10] finding that six-month-old bilinguals look increasingly less at a repeatedly presented visual stimulus than age-matched monolingual peers.

In summary, there has been much controversy over claims in the literature that bilinguals outperform monolinguals on non-verbal tasks of executive function. We could not replicate the finding that only bilingual infants can inhibit a learned behaviour. But we found that infants exposed to bilingual environments switch attention more frequently between two visual stimuli than infants exposed to monolingual environments. Infants exposed to bilingual environments may also be faster at disengaging visual attention from one stimulus in order to shift it to another. These findings are consistent with the proposal that greater variation or uncertainty in the language environment drives infants to sample (explore) more and place greater weight on novel information. However, though we argue that volatility in the bilingual environment drives infants to explore more, time spent exploring (seeking new information) may come at the expense of time spent consolidating (or exploiting) information. Therefore, we do not insist that switching attention more frequently is necessarily an 'advantage'; rather, we argue that it is an adaptation to a particular set of circumstances. We imagine that, at a particular time in development, it may benefit bilinguals to switch attention more frequently. It would be interesting to know whether these adaptations have cascading effects such as worse metacognition in early adulthood [18] or better mental health in old age [43].

Ethics. The study was approved by the Faculty Research Ethics Panel of Anglia Ruskin University and the Departmental Ethics Committee of the Department of Psychological Sciences, Birkbeck, University of London. Informed consent was obtained from the research participants' parents/guardians.

Data accessibility. The approved Stage 1 protocol has been made freely available on a public archive, the Open Science Framework (osf.io/53gh2/). The experimental stimuli, data processing scripts, and laboratory log have also been made freely available on the Open Science Framework (osf.io/53gh2/). The raw and processed data have been made freely available on the Dryad Digital Repository (https://doi.org/10.5061/dryad.3n5tb2rc6) [46].

Authors' contributions. D.D. conceived the study, designed the new experiments, planned the analyses, coordinated the study, drafted the initial registered report (for Stage 1), collected the data, analysed the data and drafted the final manuscript (for Stage 2). D.B. wrote the scripts for three of the four experiments (numbers 1, 2 and 4). J.H. processed the gap-overlap data (experiment 3). H.D. helped to conceive the study, design the experiments, draft the

initial registered report, analyse the data and critically revise the final manuscript. All authors gave final approval for publication and agree to be held accountable for the work performed therein.

Competing interests. We declare we have no competing interests.

Funding. This work was supported by the British Academy (grant no. 173426). H.D. is the Beatrice Mary Dale Research Fellow supported by Newnham College, University of Cambridge.

Acknowledgements. We would like to thank all the parents who took part in the study, without whom none of this work would have been possible. We would also like to thank Luke Mason for helping to implement experiment 3, and thank members of the ELAN laboratory (Early Learning and Neurodevelopment) for their help in recruiting and testing participants—especially Michelle Campbell-Logan, Stefania Cangemi, Emma Luck and Isabel Quiroz.

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
