## [Reviewer comments · Royal Society Open Science]

Review History

RSOS-171564.R0 (Original submission)

Review form: Reviewer 1

Is the language acceptable?

Yes

Do you have any ethical concerns with this paper?

No

Have you any concerns about statistical analyses in this paper?

Yes

Recommendation?

Major revision

Comments to the Author(s)

This Registered Report protocol presents an ambitious attempt to extend the state of the art with respect to bilingualism and cognitive control in infants prior to the onset of productive language.

My overall assessment is that this is a thorough and systematic set of studies that I would be excited to see carried out, provided my comments are addressed. Even the first experiment (the Kovacs & Mehler 2009 replication) would itself be quite a useful contribution. Despite my enthusiasm I had some specific concerns regarding the tasks, theoretical grounding, and proposed statistical analyses, detailed below, that would make me worried about going forward without seeing revisions to the protocol.

First, anticipatory looking paradigms are quite tricky. Without extensive piloting/paradigm timing customization, it can be difficult to see any anticipation at all. Many labs that use this kind of paradigm report that extensive piloting is necessary to ensure an adequate level of anticipation (without which there is no way to see differences on the manipulation of interest). It would be quite disappointing to see the results of Experiments 1 and 2 be uninterpretable due to a low proportion anticipations. I am not sure how to address this as E1 in particular is a replication study and some aspects of a failure might be due to issues in the original experiment. At a minimum, some discussion of failures to anticipate and some analyses of the proportion of anticipations seems critical for interpretability. More generally, some presentation or discussion of the age-appropriateness/validity of the paradigms being used would be useful. If the authors have pilot data it would not be inappropriate to show them in the RR to demonstrate that basic methodological issues (retention, data loss) have been piloted. (In a grant situation, I would vastly prefer such pilot data because they would signal the feasibility of measuring the effect of interest).

Theoretically, I found that the introduction did not seem to take seriously the possibility that bilingualism findings with preverbal infants might be artifactual or incorrect. There are at least four theoretical possibilities in this space (logically):

1. bilingual CC advantage caused by production (attributed to Green),
2. " caused by home environmental (attributed to D'Souza & D'Souza),
3. bilingual CC advantage due to other causes, and
4. No bilingual CC advantage

The proposed experiments are informative regarding this space, with failures to observe differences consistent with both 1 and 4 and successes consistent with 2 or 3. I thought that a little more care in navigating this theoretical space was warranted.

On the topic of participants: Why not use a standardized instrument for gathering language background? Self report of percentages is problematic and many folks have worked on this extensively. There are several well-validated instruments, e.g.:

- Cattani, A., Abbot-Smith, K., Farag, R., Krott, A., Arreckx, F., Dennis, I., & Floccia, C. (2014). How much exposure to English is necessary for a bilingual toddler to perform like a monolingual peer in language tests? *International Journal of Language and Communication Disorders*.
- Li, P., Zhang, F., Tsai, E., & Puls, B. (2014). Language history questionnaire (LHQ 2.0): A new dynamic web-based research tool. *Bilingualism: Language and Cognition*.

Likely the background of the infants tested will be one of the major interpretive points in parsing the results - it is critical then to have a strong characterization of language background. The current short questionnaire is subject to biases in reporting (self report of percentages is quite noisy) and quite brief.

Other comments, mostly on analysis:

* Will the authors confirm with parents that children are not yet producing words? This will be true for the majority of children, but surprisingly, there are some children in the 8-9 month range that have produced a first word.

* K&M2009 analysis: "To test this hypothesis, we will measure proportion of correct anticipatory looks averaged over the final three trials of the post-switch phase. The proportional data will be arcsine transformed and analysed using an independent samples t-test." A more sensitive

analysis would be logistic mixed effects regression, which could account for trial order and would not require a transform. I'd strongly recommend that the authors consider that model as the primary analysis (unless they want to designate a "replication analysis" and preregister that too - the original authors appear to have used an f-test with no transform).

* Do the authors want to take further advantage of trial order and model the trend over more than 3 trials (increasing power), as in the linear trends shown in K&M2009's original paper?

* "We will run independent samples t-tests to check that the two groups did not significantly differ from each another on age, gender, and parents' SES." Another, potentially more powerful, possibility would be to control for these in planned analyses, e.g. the model described above.

* "For each experimental task, we will report any statistically significant difference between groups on the number of trials that data were provided." Wasn't sure what this meant.

Review form: Reviewer 2

Is the language acceptable?

Yes

Do you have any ethical concerns with this paper?

No

Have you any concerns about statistical analyses in this paper?

No

Recommendation?

Accept with minor revision

Comments to the Author(s)

This paper proposes to carry out a series of eye-tracking experiments with a cohort of 102 7- to 9-month-olds (half bilinguals, half monolinguals), to revisit the claims that bilinguals develop a cognitive advantage.

The manuscript is very well written overall and the ideas are very sound. I am particularly eager to see a large-scale replication of Kovacs and Mehler (2009). Indeed the demonstration that 7-month-old bilinguals had better inhibitory skills than monolinguals has strong implications, and as such, a replication is very welcome.

I am also very interested in the extension of the investigation to other aspects of cognition such as attention shifting (Exp 3), but a bit more sceptical about the claims that bilingual children would be sampling less visual information (Exp 2 and 4). I am not convinced by the logic according which more complex environments would lead to impoverished representations of the world. For example, in the acquisition of morphosyntax, kids who are acquiring more complex morphologies are not delayed as compared to those who are learning simpler ones, they do reach an equal level of expertise within the same timeframe. However I would be very interested to see the outcomes of exp 2 and 4.

One caveat is the questionnaire used to calculate exposure: asking parents their evaluation of the amount of time each member of the family speaks language A or B does not sound like the most accurate way to estimate exposure - and exposure is a key factor here as it helps determine whether a child is in group A or B. There are a few available questionnaires out there which are more accurate (and not long at all, see for example the Plymouth LEQ which is online at <http://www.psy.plymouth.ac.uk/babylab/languageexposure.html>)

Review form: Reviewer 3

Is the language acceptable?

Yes

Do you have any ethical concerns with this paper?

No

Have you any concerns about statistical analyses in this paper?

Yes

Recommendation?

Major revision

Comments to the Author(s)

This is in general a very interesting study that will test an alternative account of bilingual cognitive advantage in infants. There is currently a hot debate about the bilingual cognitive advantage, and the infant data are puzzling. Thus, the study is very timely.

The introduction, logic and rationale seem good. I do not have any comments or questions about them. I do, however, have some questions about the materials and methods. Some of these are just clarification questions, others more critical. Importantly, I am not convinced (yet) that Experiment 4 tests what it is meant to do.

p. 4, line 20. Participants: Please state how you will measure parents' socioeconomic status.

p.4, line 24. It is stated that results of a particular infant are included into the analysis of a particular experiment when the infant has at least 75% usable eye tracking data for that experiment. But what will you do if an infant produces 75% good data overall, but only 1 or 2 valid trials out of the small number of trials you actually analyse (these are never more than 3 trials). It seems to me that these critical trials need to be valid trials as well.

p.4, line 29: It is stated that if not all infants produce enough valid trials for all experiments, that more infants are tested so that there will be 51 infants of each participants group in all experiments. This will mean that some infants won't take part in all experiments. Nevertheless, it seems to me useful to analyse whether performance on the 4 experiments is correlated. In other words, if a particular infant shows superior performance in one experiment, is he likely to do this also on the other experiments? Otherwise it might be that different bilingual infants might drive a bilingual superiority effect in different experiments. If so, one cannot generalise to bilingual infants as a population.

p. 4, line 42: what is the purpose of the camera that is mounted directly above the horizontal midpoint of the screen? It is stated that this will monitor and record infant behaviour. Why is this important? These recordings do not seem to be analysed or used in any way.

p. 6, Experiment 1, coding and analysis:

It is stated that if an infant looks at both the left and right side of the screen during the critical period, that the longer look will be coded. Do you mean that if an infant looks 51% of the time to the left side and 49% of the time to the right side, that you will treat their look as if they were looking only to the left side? That does not seem to be right as it does not account for the actual looking behaviour. I would suggest to use a stricter criterion. For instance, that the longer look has to be at least 75% of the looking behaviour.

Related to that, why are looks coded as right, middle and left, and not whether the infants look to the white boxes where the rewards will appear? (see also other experiments) Left and right seems a very rough measure if you have an eye tracker that provides more precise information.

p. 6, line 38: In order to be sure that infants have actually learned to anticipate the reward in the pre-switch phase, you will need to also analyse the proportion of correct anticipatory looks averaged, for instance, over the final three of the pre-switch phase.

p.6 line 38: Related to my question above: What if not all three final post-switch trials are valid trials? Will you take the 1 or 2 valid trials instead? That seems a very small number of trials to draw a strong conclusion from.

p. 6, line 41: Here and for at least one other experiment, it is stated that infants looking behaviour will be compared against chance. Please state what exactly is a chance behaviour. Do you mean by chance a proportion of 0.5 to the correct side? I would find this problematic. Looking left or right are not the only options that the infant has. They can also look to the middle of the screen or not at all to the screen.

p. 7, Experiment 2, figure 2:

Can you please also add the figures of the elephant that you will use as stimuli.

p. 8, line 14: two trials seem very little to draw conclusions from (especially if one of them is not a valid trial). Are two trials enough?

Experiment 3

p. 9: where does the reward appear?

p. 9, line 18: how do you code a trial where the infant shifts their gaze away from the central fixation stimulus, but not towards the target?

p. 9, line 23: why does one need to subtract RTs in the baseline condition from RTs in the overlap condition? Why can one not simply use the RT in the overlap condition as a measure of failure to disengage?

P. 9, line 25: I do not understand why the gap condition is a control for fast saccading. Please explain.

Experiment 4:

-P. 9, line 38: state here what the line drawings show. It would be useful to have some examples of how the line drawings look when changing over the 15 trials, especially of the critical trials that are analysed. Do the critical trials actually show meaningful objects?

p. 9, line 52: what is the purpose of the background music?

p. 10, line 3: it is stated that the number of switches are measured. Is this on trials 2 and 3, or all 15 trials?

p. 10, line 4: please define 'familiar' stimulus and 'novel stimulus/side'. I don't understand.

p. 10, line 12: why do you look at trials 6 and 7?

p. 10, line 4/5. It is stated that bilinguals are expected to switch more frequently because they err on the side of exploration compared to monolinguals. But is it not also possible that frequent switches mean that an infant realises the change and is interested in what is going on at both sides? Checking whether both sides are changing or only one side? And why would monolinguals rather look at the familiar (=non changing picture) when they want to build

detailed models? And how can you tell that both groups of infants can equally discriminate between the novel and familiar picture? And why is looking at the changing picture evidence of not remembering the change? If bilinguals explore a lot, they will notice the change at trials 6 and 7 because these will look very different from what they had seen at the beginning of the experiment. In sum, I am afraid I am not convinced that this experiment actually measures what it is supposed to measure.

Decision letter (RSOS-171564.R0)

27-Nov-2017

Dear Dr D'Souza,

The Editors assigned to your Stage 1 Registered Report ("Is exposure enough? The effects of bilingual environments on infant cognitive development") have now received comments from reviewers. We would like you to revise your paper in accordance with the referee and editors suggestions which can be found below (not including confidential reports to the Editor). Please note this decision does not guarantee eventual acceptance.

Your manuscript is likely to be sent back to one or more of the original reviewers for assessment. If the original reviewers are not available we may invite new reviewers.

Please note that Royal Society Open Science will introduce article processing charges for all new submissions received from 1 January 2018. Registered Reports submitted and accepted after this date will ONLY be subject to a charge if they subsequently progress to and are accepted as Stage 2 Registered Reports. If your manuscript is submitted and accepted for publication after 1 January 2018 (i.e. as a full Stage 2 Registered Report), you will be asked to pay the article processing charge, unless you request a waiver and this is approved by Royal Society Publishing. You can find out more about the charges at <http://rsos.royalsocietypublishing.org/page/charges>. Should you have any queries, please contact openscience@royalsociety.org.

on behalf of Chris Chambers (Registered Reports Editor, Royal Society Open Science)
openscience@royalsociety.org

Editor Comments to Author:

Three expert reviewers have appraised the manuscript. The reviews are overall positive but raise critical issues that span the full range of Stage 1 review criteria. These will need to be addressed comprehensively in revision to achieve IPA.

Reviewer 1 is generally enthusiastic about the proposal but requests greater methodological detail/justification and discussion of theory. One major issue raised by this reviewer is the risk that some of the proposed hypotheses may turn out to be untestable due to a low proportion of anticipatory looks. Pilot data here seems sensible to consider, and at the least, a minimum level of anticipatory looks should be specified in the protocol in order to satisfy Stage 1 Criterion 6: "Whether the authors have considered sufficient outcome-neutral conditions (e.g. absence of floor or ceiling effects; positive controls; other quality checks) for ensuring that the results obtained are able to test the stated hypotheses." Reviewer 1 also recommends alternative mixed effects analyses, and greater consideration in the introduction the possibility that bilingual advantage may be artefactual. Reviewer 2 is overall positive but raises questions about the rationale of experiment 2 and 4 and suggests an alternative measure of language exposure. Reviewer 3 shares Reviewer 2's skepticism about experiment 4. The reviewer also questions the level of methodological detail and rationale across multiple experiments, including exclusion criteria, coding and analysis plans.

Comments to Author:

Reviewer: 1

Comments to the Author(s)

This Registered Report protocol presents an ambitious attempt to extend the state of the art with respect to bilingualism and cognitive control in infants prior to the onset of productive language. My overall assessment is that this is a thorough and systematic set of studies that I would be excited to see carried out, provided my comments are addressed. Even the first experiment (the Kovacs & Mehler 2009 replication) would itself be quite a useful contribution. Despite my enthusiasm I had some specific concerns regarding the tasks, theoretical grounding, and proposed statistical analyses, detailed below, that would make me worried about going forward without seeing revisions to the protocol.

First, anticipatory looking paradigms are quite tricky. Without extensive piloting/paradigm timing customization, it can be difficult to see any anticipation at all. Many labs that use this kind of paradigm report that extensive piloting is necessary to ensure an adequate level of anticipation (without which there is no way to see differences on the manipulation of interest). It would be quite disappointing to see the results of Experiments 1 and 2 be uninterpretable due to a low proportion of anticipations. I am not sure how to address this as E1 in particular is a replication study and some aspects of a failure might be due to issues in the original experiment. At a minimum, some discussion of failures to anticipate and some analyses of the proportion of anticipations seems critical for interpretability. More generally, some presentation or discussion of the age-appropriateness/validity of the paradigms being used would be useful. If the authors have pilot data it would not be inappropriate to show them in the RR to demonstrate that basic methodological issues (retention, data loss) have been piloted. (In a grant situation, I would vastly prefer such pilot data because they would signal the feasibility of measuring the effect of interest).

Theoretically, I found that the introduction did not seem to take seriously the possibility that bilingualism findings with preverbal infants might be artifactual or incorrect. There are at least four theoretical possibilities in this space (logically):

1. bilingual CC advantage caused by production (attributed to Green),
2. " caused by home environmental (attributed to D'Souza & D'Souza),
3. bilingual CC advantage due to other causes, and
4. No bilingual CC advantage

The proposed experiments are informative regarding this space, with failures to observe differences consistent with both 1 and 4 and successes consistent with 2 or 3. I thought that a little more care in navigating this theoretical space was warranted.

On the topic of participants: Why not use a standardized instrument for gathering language background? Self report of percentages is problematic and many folks have worked on this extensively. There are several well-validated instruments, e.g.:

- Cattani, A., Abbot-Smith, K., Farag, R., Krott, A., Arreckx, F., Dennis, I., & Floccia, C. (2014). How much exposure to English is necessary for a bilingual toddler to perform like a monolingual peer in language tests? *International Journal of Language and Communication Disorders*.
- Li, P., Zhang, F., Tsai, E., & Puls, B. (2014). Language history questionnaire (LHQ 2.0): A new dynamic web-based research tool. *Bilingualism: Language and Cognition*.

Likely the background of the infants tested will be one of the major interpretive points in parsing the results - it is critical then to have a strong characterization of language background. The current short questionnaire is subject to biases in reporting (self report of percentages is quite noisy) and quite brief.

Other comments, mostly on analysis:

* Will the authors confirm with parents that children are not yet producing words? This will be true for the majority of children, but surprisingly, there are some children in the 8-9 month range that have produced a first word.

* K&M2009 analysis: "To test this hypothesis, we will measure proportion of correct anticipatory looks averaged over the final three trials of the post-switch phase. The proportional data will be arcsine transformed and analysed using an independent samples t-test." A more sensitive analysis would be logistic mixed effects regression, which could account for trial order and would not require a transform. I'd strongly recommend that the authors consider that model as the primary analysis (unless they want to designate a "replication analysis" and preregister that too - the original authors appear to have used an f-test with no transform).

* Do the authors want to take further advantage of trial order and model the trend over more than 3 trials (increasing power), as in the linear trends shown in K&M2009's original paper?

* "We will run independent samples t-tests to check that the two groups did not significantly differ from each another on age, gender, and parents' SES." Another, potentially more powerful, possibility would be to control for these in planned analyses, e.g. the model described above.

* "For each experimental task, we will report any statistically significant difference between groups on the number of trials that data were provided." Wasn't sure what this meant.

Reviewer: 2

Comments to the Author(s)

This paper proposes to carry out a series of eye-tracking experiments with a cohort of 102 7- to 9-month-olds (half bilinguals, half monolinguals), to revisit the claims that bilinguals develop a cognitive advantage.

The manuscript is very well written overall and the ideas are very sound. I am particularly eager to see a large-scale replication of Kovacs and Mehler (2009). Indeed the demonstration that 7-month-old bilinguals had better inhibitory skills than monolinguals has strong implications, and as such, a replication is very welcome.

I am also very interested in the extension of the investigation to other aspects of cognition such as attention shifting (Exp 3), but a bit more sceptical about the claims that bilingual children would be sampling less visual information (Exp 2 and 4). I am not convinced by the logic according which more complex environments would lead to impoverished representations of the world. For example, in the acquisition of morphosyntax, kids who are acquiring more complex morphologies are not delayed as compared to those who are learning simpler ones, they do reach an equal level of expertise within the same timeframe. However I would be very interested to see the outcomes of exp 2 and 4.

One caveat is the questionnaire used to calculate exposure: asking parents their evaluation of the amount of time each member of the family speaks language A or B does not sound like the most accurate way to estimate exposure – and exposure is a key factor here as it helps determine whether a child is in group A or B. There are a few available questionnaires out there which are more accurate (and not long at all, see for example the Plymouth LEQ which is online at <http://www.psy.plymouth.ac.uk/babylab/languageexposure.html>)

Reviewer: 3

Comments to the Author(s)

This is in general a very interesting study that will test an alternative account of bilingual cognitive advantage in infants. There is currently a hot debate about the bilingual cognitive advantage, and the infant data are puzzling. Thus, the study is very timely.

The introduction, logic and rationale seem good. I do not have any comments or questions about them. I do, however, have some questions about the materials and methods. Some of these are just clarification questions, others more critical. Importantly, I am not convinced (yet) that Experiment 4 tests what it is meant to do.

p. 4, line 20. Participants: Please state how you will measure parents' socioeconomic status.

p.4, line 24. It is stated that results of a particular infant are included into the analysis of a particular experiment when the infant has at least 75% usable eye tracking data for that experiment. But what will you do if an infant produces 75% good data overall, but only 1 or 2 valid trials out of the small number of trials you actually analyse (these are never more than 3 trials). It seems to me that these critical trials need to be valid trials as well.

p.4, line 29: It is stated that if not all infants produce enough valid trials for all experiments, that more infants are tested so that there will be 51 infants of each participants group in all experiments. This will mean that some infants won't take part in all experiments. Nevertheless, it seems to me useful to analyse whether performance on the 4 experiments is correlated. In other words, if a particular infant shows superior performance in one experiment, is he likely to do this also on the other experiments? Otherwise it might be that different bilingual infants might drive a bilingual superiority effect in different experiments. If so, one cannot generalise to bilingual infants as a population.

p. 4, line 42: what is the purpose of the camera that is mounted directly above the horizontal midpoint of the screen? It is stated that this will monitor and record infant behaviour. Why is this important? These recordings do not seem to be analysed or used in any way.

p. 6, Experiment 1, coding and analysis:

It is stated that if an infant looks at both the left and right side of the screen during the critical period, that the longer look will be coded. Do you mean that if an infant looks 51% of the time to

the left side and 49% of the time to the right side, that you will treat their look as if they were looking only to the left side? That does not seem to be right as it does not account for the actual looking behaviour. I would suggest to use a stricter criterion. For instance, that the longer look has to be at least 75% of the looking behaviour.

Related to that, why are looks coded as right, middle and left, and not whether the infants look to the white boxes where the rewards will appear? (see also other experiments) Left and right seems a very rough measure if you have an eye tracker that provides more precise information.

p. 6, line 38: In order to be sure that infants have actually learned to anticipate the reward in the pre-switch phase, you will need to also analyse the proportion of correct anticipatory looks averaged, for instance, over the final three of the pre-switch phase.

p.6 line 38: Related to my question above: What if not all three final post-switch trials are valid trials? Will you take the 1 or 2 valid trials instead? That seems a very small number of trials to draw a strong conclusion from.

p. 6, line 41: Here and for at least one other experiment, it is stated that infants looking behaviour will be compared against chance. Please state what exactly is a chance behaviour. Do you mean by chance a proportion of 0.5 to the correct side? I would find this problematic. Looking left or right are not the only options that the infant has. They can also look to the middle of the screen or not at all to the screen.

p. 7, Experiment 2, figure 2:

Can you please also add the figures of the elephant that you will use as stimuli.

p. 8, line 14: two trials seem very little to draw conclusions from (especially if one of them is not a valid trial). Are two trials enough?

Experiment 3

p. 9: where does the reward appear?

p. 9, line 18: how do you code a trial where the infant shifts their gaze away from the central fixation stimulus, but not towards the target?

p. 9, line 23: why does one need to subtract RTs in the baseline condition from RTs in the overlap condition? Why can one not simply use the RT in the overlap condition as a measure of failure to disengage?

P. 9, line 25: I do not understand why the gap condition is a control for fast saccading. Please explain.

Experiment 4:

-P. 9, line 38: state here what the line drawings show. It would be useful to have some examples of how the line drawings look when changing over the 15 trials, especially of the critical trials that are analysed. Do the critical trials actually show meaningful objects?

p. 9, line 52: what is the purpose of the background music?

p. 10, line 3: it is stated that the number of switches are measured. Is this on trials 2 and 3, or all 15 trials?

p. 10, line 4: please define 'familiar' stimulus and 'novel stimulus/side'. I don't understand.

p. 10, line 12: why do you look at trials 6 and 7?

p. 10, line 4/5. It is stated that bilinguals are expected to switch more frequently because they err on the side of exploration compared to monolinguals. But is it not also possible that frequent switches mean that an infant realises the change and is interested in what is going on at both sides? Checking whether both sides are changing or only one side? And why would monolinguals rather look at the familiar (=non changing picture) when they want to build detailed models? And how can you tell that both groups of infants can equally discriminate between the novel and familiar picture? And why is looking at the changing picture evidence of not remembering the change? If bilinguals explore a lot, they will notice the change at trials 6 and 7 because these will look very different from what they had seen at the beginning of the experiment. In sum, I am afraid I am not convinced that this experiment actually measures what it is supposed to measure.

Author's Response to Decision Letter for (RSOS-171564.R0)

See Appendix A.

RSOS-180045.R0

Review form: Reviewer 1

Is the language acceptable?

Yes

Do you have any ethical concerns with this paper?

No

Have you any concerns about statistical analyses in this paper?

No

Recommendation?

Accept in principle

Comments to the Author(s)

The authors have submitted a responsive revision to their RR protocol and I am now enthusiastic to see the results of their ambitious study.

Review form: Reviewer 2

Is the language acceptable?

Yes

Do you have any ethical concerns with this paper?

No

Have you any concerns about statistical analyses in this paper?

No

Recommendation?

Accept in principle

Comments to the Author(s)

None

Review form: Reviewer 3

Is the language acceptable?

Yes

Do you have any ethical concerns with this paper?

No

Have you any concerns about statistical analyses in this paper?

No

Recommendation?

Accept with minor revision

Comments to the Author(s)

The authors have addressed most of my comments and I am generally happy with the study and the rationale behind the experiments. I do have one very minor comment and still some reservations about experiment 4 and the predicted outcomes and their interpretations.

The authors provide a lot of explanations about the processes in Experiment 4 that make a lot of sense and that I fully agree with. But I believe that the logic, the potential results and their interpretation have still not been fully thought through.

I agree that more frequent switching between the two stimuli means more exploration than less frequent switching. And the pilot data nicely suggest that the authors will find that bilingual children generally switch more frequently and therefore seem to be less eager in gaining detailed information about the stimuli than monolinguals.

I also agree that if the monolingual (or bilingual) infants as a group look longer at one of the two stimuli (changing or non-changing) that they seem to notice a difference between the stimuli. But as the authors explain themselves in the reply to my previous concern, if a group of children do not look longer at one of the stimuli, then one cannot draw any firm conclusion. If they found this for bilingual children (as indicated in the pilot), then one cannot conclude that the bilingual children did not see that one of the stimuli changed. This also means that an investigation into the timing of when a group looks longer at one stimulus compared to the other stimulus is not a reliable indication of when the change was noticed. Bilinguals might keep comparing the stimuli despite or maybe even because of noticing the change, while monolinguals (who are more interested in building up detailed representations) might stop comparing.

In conclusion, I think that more explorative behaviour versus more focussed looks to a particular stimulus can tell us something about the different behaviour of monolingual and bilingual infants, but it might not tell us anything about whether the infants noticed the change (in the case that the looks are not longer to one of the two stimuli) or whether one group of infants notices the change earlier.

Minor points:

Pilot data for experiment 4: it is not clear whether the monolingual children looked more to the familiar of changing stimulus.

p. 11, line 27: do the authors mean they will test whether monolinguals will look longer at the familiar stimulus than predicted by chance (not the novel one)?

Could the authors please add a note in the text about why the background music was used in experiment 4 (they explained this in the response to my question, but not in the document).

Decision letter (RSOS-180045.R0)

30-Jan-2018

Dear Dr D'Souza

On behalf of the Editors, I am pleased to inform you that your Manuscript RSOS-180045 entitled "Is mere exposure enough? The effects of bilingual environments on infant cognitive development" has been accepted in principle for publication in Royal Society Open Science subject to minor revision in accordance with the referee and editor suggestions. Please find their comments at the end of this email.

The reviewers and handling editors have recommended in principle acceptance pending minor revisions to your manuscript. Therefore, I invite you to respond to the comments and revise your manuscript.

Please you submit the revised version of your manuscript within 7 days (i.e. by the 07-Feb-2018). If you do not think you will be able to meet this date please let me know immediately.

Full author guidelines can be found here

<http://rsos.royalsocietypublishing.org/content/registered-reports>.

Please note that Royal Society Open Science will introduce article processing charges for all new submissions received from 1 January 2018. Registered Reports submitted and accepted after this date will ONLY be subject to a charge if they subsequently progress to and are accepted as Stage 2 Registered Reports. If your manuscript is submitted and accepted for publication after 1 January 2018 (i.e. as a full Stage 2 Registered Report), you will be asked to pay the article processing charge, unless you request a waiver and this is approved by Royal Society Publishing. You can find out more about the charges at <http://rsos.royalsocietypublishing.org/page/charges>. Should you have any queries, please contact opscience@royalsociety.org.

on behalf of Professor Chris Chambers (Subject Editor, Royal Society Open Science)
openscience@royalsociety.org

Associate Editor Comments to Author (Professor Chris Chambers):

The three expert reviewers have responded positively to the revised manuscript, with two recommending IPA and one suggesting further minor revision. Please attend carefully to these remaining comments from Reviewer 3, especially concerning the potential limitations of Experiment 4. Provided that any further revisions are judged to adequately address this reviewer's points, IPA should be forthcoming without requiring further in-depth Stage 1 review.

Reviewer comments to Author:

Reviewer: 1

Comments to the Author(s)

The authors have submitted a responsive revision to their RR protocol and I am now enthusiastic to see the results of their ambitious study.

Reviewer: 3

Comments to the Author(s)

The authors have addressed most of my comments and I am generally happy with the study and the rationale behind the experiments. I do have one very minor comment and still some reservations about experiment 4 and the predicted outcomes and their interpretations.

The authors provide a lot of explanations about the processes in Experiment 4 that make a lot of sense and that I fully agree with. But I believe that the logic, the potential results and their interpretation have still not been fully thought through.

I agree that more frequent switching between the two stimuli means more exploration than less frequent switching. And the pilot data nicely suggest that the authors will find that bilingual children generally switch more frequently and therefore seem to be less eager in gaining detailed information about the stimuli than monolinguals.

I also agree that if the monolingual (or bilingual) infants as a group look longer at one of the two stimuli (changing or non-changing) that they seem to notice a difference between the stimuli. But as the authors explain themselves in the reply to my previous concern, if a group of children do not look longer at one of the stimuli, then one cannot draw any firm conclusion. If they found this for bilingual children (as indicated in the pilot), then one cannot conclude that the bilingual children did not see that one of the stimuli changed. This also means that an investigation into the timing of when a group looks longer at one stimulus compared to the other stimulus is not a reliable indication of when the change was noticed. Bilinguals might keep comparing the stimuli

despite or maybe even because of noticing the change, while monolinguals (who are more interested in building up detailed representations) might stop comparing.

In conclusion, I think that more explorative behaviour versus more focussed looks to a particular stimulus can tell us something about the different behaviour of monolingual and bilingual infants, but it might not tell us anything about whether the infants noticed the change (in the case that the looks are not longer to one of the two stimuli) or whether one group of infants notices the change earlier.

Minor points:

pilot data for experiment 4: it is not clear whether the monolingual children looked more to the familiar of changing stimulus.

p. 11, line 27: do the authors mean they will test whether monolinguals will look longer at the familiar stimulus than predicted by chance (not the novel one)?

Could the authors please add a note in the text about why the background music was used in experiment 4 (they explained this in the response to my question, but not in the document).

Reviewer: 2

Comments to the Author(s)

None

Author's Response to Decision Letter for (RSOS-180045.R0)

See Appendix B.

Decision letter (RSOS-180191.R0)

07-Feb-2018

Dear Dr D'Souza

On behalf of the Editor, I am pleased to inform you that your Manuscript RSOS-180191 entitled "Is mere exposure enough? The effects of bilingual environments on infant cognitive development" has been accepted in principle for publication in Royal Society Open Science.

You may now progress to Stage 2 and complete the study as approved. Before commencing data collection we ask that you:

- 1) Update the journal office as to the anticipated completion date of your study.
- 2) Register your approved protocol on the Open Science Framework (<https://osf.io/>) or other recognised repository, either publicly or privately under embargo until submission of the Stage 2 manuscript. Please note that a time-stamped, independent registration of the protocol is mandatory under journal policy, and manuscripts that do not conform to this requirement cannot be considered at Stage 2. The protocol should be registered unchanged from its current approved state, with the time-stamp preceding implementation of the approved study design.

Following completion of your study, we invite you to resubmit your paper for peer review as a Stage 2 Registered Report. Please note that your manuscript can still be rejected for publication at Stage 2 if the Editors consider any of the following conditions to be met:

- The results were unable to test the authors' proposed hypotheses by failing to meet the approved outcome-neutral criteria.
- The authors altered the Introduction, rationale, or hypotheses, as approved in the Stage 1 submission.
- The authors failed to adhere closely to the registered experimental procedures. Please note that any deviations from the approved experimental procedures must be communicated to the editor immediately for approval, and prior to the completion of data collection. Failure to do so can result in revocation of in-principle acceptance and rejection at Stage 2 (see complete guidelines for further information).
- Any post-hoc (unregistered) analyses were either unjustified, insufficiently caveated, or overly dominant in shaping the authors' conclusions.
- The authors' conclusions were not justified given the data obtained.

We encourage you to read the complete guidelines for authors concerning Stage 2 submissions at <http://rsos.royalsocietypublishing.org/content/registered-reports>. Please especially note the requirements for data sharing, reporting the URL of the independently registered protocol, and that withdrawing your manuscript will result in publication of a Withdrawn Registration.

Once again, thank you for submitting your manuscript to Royal Society Open Science and we look forward to receiving your Stage 2 submission. If you have any questions at all, please do not hesitate to get in touch. We look forward to hearing from you shortly with the anticipated submission date for your Stage 2 manuscript.

Kind regards,

Alice Power
Editorial Coordinator
Royal Society Open Science
openscience@royalsociety.org

on behalf of Professor Chris Chambers (Registered Reports Editor, Royal Society Open Science)
openscience@royalsociety.org

Author's Response to Decision Letter for (RSOS-180191.R0)

See Appendix C.

RSOS-180191.R1 (Revision)

Review form: Reviewer 1

Is the manuscript scientifically sound in its present form?

Yes

Are the interpretations and conclusions justified by the results?

Yes

Is the language acceptable?

Yes

Do you have any ethical concerns with this paper?

No

Have you any concerns about statistical analyses in this paper?

No

Recommendation?

Accept with minor revision

Comments to the Author(s)

I have read over this Stage 2 registered report and am satisfied that it meets the standards set out in the Stage 1 report.

I did not see code or materials in the linked OSF repository, only lab logs and preregistration documents. There was also no Dryad link. I thus could not verify that these files were shared.

As the authors write in their lab log, "Experimental tasks 2 and 4 were too difficult for the infants and need to be completely redesigned for this age group." This is a real risk of the RR format - the experiments have limited evidential value for reasons that were difficult for both the authors and the reviewers to foresee. Despite this, I found the authors' discussion of the issues generally appropriate.

I would recommend some minor revision of the discussion section. First, the initial paragraph should be broken up into several distinct paragraphs with different points - at least a clear summary of the current finding and a discussion with respect to related work. Second, the oblique "data from one lab" comment felt odd here because the finding that the authors fail to replicate is from Kovacs & Mehler, not (what I assume to be) the Bialystok group.

Minor:

- "This demonstrates the importance of registered reports" - I don't think this line is warranted in the conclusion. RRs do not guarantee comparable procedures or analyses across groups.

Review form: Reviewer 3

Is the manuscript scientifically sound in its present form?

Yes

Are the interpretations and conclusions justified by the results?

Yes

Is the language acceptable?

Yes

Do you have any ethical concerns with this paper?

No

Have you any concerns about statistical analyses in this paper?

No

Recommendation?

Accept with minor revision

Comments to the Author(s)

1. Are the data able to test the authors' proposed hypotheses by passing the approved outcome-neutral criteria (such as absence of floor and ceiling effects or success of positive controls)?
The data analysis follows what had been proposed. In addition, some useful explorative analyses were conducted. Those make a lot of sense and are doing justice to the data.

Having said this, I have a comment on the analyses of Experiments 1 and 2. I noticed that anticipatory looks are defined as saccades within 1s starting 150ms after cue offset. It seems that this is what Kovac and Mehler (2009) did. But I find the start of this period quite restricted. For instance, in Experiment 1, the last cue shape is presented for 0.8s. Isn't it possible that infants move their eyes before that last cue shape disappears? In other words, would an earlier start of that period make a difference to the results? I would have thought 150ms after the onset (not offset) of the last cue would be more appropriate.

2. Are the introduction, rationale and stated hypotheses the same as the approved Stage 1 submission?

Yes, these are the same.

3. Did the authors adhere precisely to the registered experimental procedures?

Yes, they generally did, apart from some very small changes that the authors have marked in red. But these changes do not change the hypotheses or the conclusions.

To note is that procedure and analysis sections are written in future tense. I am not familiar with the policy of the journal, but I would suggest to change them to past tense.

I also would find it helpful if the results of each of the four experiments could follow directly the method section. It is otherwise difficult to remember the details of the experiments.

I would also move the comparison of the socioeconomic status and age of the two participant groups to the participant section.

4. Where applicable, are any unregistered exploratory statistical analyses justified, methodologically sound, and informative?

The additional analyses are very useful and appear sound to me.

5. Whether the authors' conclusions are justified given the data

I generally agree with the conclusions. But I have some further questions and suggestions.

First, it is not entirely clear how all the results are interpreted. For that it would be useful to start the discussion section with a summary of the findings and then make sure that all findings are discussed.

Results of experiment 4 (p. 21):

I don't think that the results of the analysis of the proportion of looks to the changing stimuli necessarily mean that the task was too difficult. It could be that infants notice the change, but lose interest. It would therefore be useful to see a graph for this analysis. (see also comment below about graphs for non-significant results). Btw, I do not understand the comment 'so we cannot conclude that the difference is real' (line 53). Which difference do the authors refer to? The difference of length of looks at the familiar objects? I don't think this analysis diminishes this outcome.

Relation of findings to similar studies:

While there is a comparison of the Kovacs and Mehler (2009) study and other studies (including the present one), I find this unsatisfying. What exactly is different between the current study and the one(s) that find different results? And could one analyse the current study in a way more closely to what these other studies did? For instance, is the method of the current study exactly the same as that in Kovacs and Mehler (2009), apart from the sample size and the 'engaging stimuli'? (Btw, does the latter refer to the cue or the reward stimulus). Also, is the coding exactly the same?

Minor comments:

- Please indicate in the figure captions which experiment they belong to.
- Results of non-significant findings (e.g. Experiment 2): a graph showing the results would be very helpful for the reader.
- P. 10, line 29: change "periphe ral" to "peripheral"
- First paragraph of discussion (p. 22) is too long. Split up into smaller meaningful units.

Decision letter (RSOS-180191.R1)

02-Dec-2019

Dear Dr D'Souza:

On behalf of the Editor, I am pleased to inform you that your Stage 2 Registered Report RSOS-180191.R1 entitled "Is mere exposure enough? The effects of bilingual environments on infant cognitive development" has been deemed suitable for publication in Royal Society Open Science subject to minor revision in accordance with the referee suggestions. Please find the referees' comments at the end of this email.

The reviewers and Subject Editor have recommended publication, but also suggest some minor revisions to your manuscript. Therefore, I invite you to respond to the comments and revise your manuscript.

Please also ensure that all the below editorial sections are included where appropriate -- if any section is not applicable to your manuscript, please can we ask you to nevertheless include the heading, but explicitly state that the heading is inapplicable. An example of these sections is attached with this email.

- Ethics statement

- Data accessibility

It is a condition of publication that all supporting data are made available either as supplementary information or preferably in a suitable permanent repository. The data accessibility section should state where the article's supporting data can be accessed. This section should also include details, where possible of where to access other relevant research materials such as statistical tools, protocols, software etc can be accessed. If the data has been deposited in an external repository this section should list the database, accession number and link to the DOI for all data from the article that has been made publicly available. Data sets that have been

deposited in an external repository and have a DOI should also be appropriately cited in the manuscript and included in the reference list.

If you wish to submit your supporting data or code to Dryad (<http://datadryad.org/>), or modify your current submission to dryad, please use the following link:
[http://datadryad.org/submit?journalID=RSOS&manu=\(Document not available\)](http://datadryad.org/submit?journalID=RSOS&manu=(Document not available))

- **Competing interests**

- **Authors' contributions**

- **Acknowledgements**

- **Funding statement**

- 1) A text file of the manuscript (tex, txt, rtf, docx or doc), references, tables (including captions) and figure captions. Do not upload a PDF as your "Main Document".
- 2) A separate electronic file of each figure (EPS or print-quality PDF preferred (either format should be produced directly from original creation package), or original software format)
- 3) Included a 100 word media summary of your paper when requested at submission. Please ensure you have entered correct contact details (email, institution and telephone) in your user account

- 4) Included the raw data to support the claims made in your paper. You can either include your data as electronic supplementary material or upload to a repository and include the relevant doi within your manuscript
- 5) All supplementary materials accompanying an accepted article will be treated as in their final form. Note that the Royal Society will neither edit nor typeset supplementary material and it will be hosted as provided. Please ensure that the supplementary material includes the paper details where possible (authors, article title, journal name).

Please note that Royal Society Open Science will introduce article processing charges for all new submissions received from 1 January 2018. Registered Reports submitted and accepted after this date will ONLY be subject to a charge if they subsequently progress to and are accepted as Stage 2 Registered Reports. If your manuscript is submitted and accepted for publication after 1 January 2018 (i.e. as a full Stage 2 Registered Report), you will be asked to pay the article processing charge, unless you request a waiver and this is approved by Royal Society Publishing. You can find out more about the charges at <https://royalsocietypublishing.org/rsos/charges>. Should you have any queries, please contact openscience@royalsociety.org.

on behalf of Professor Chris Chambers
(Registered Reports Editor, Royal Society Open Science)
openscience@royalsociety.org

Associate Editor Comments to Author (Professor Chris Chambers):

Two of the expert reviewers who approved the manuscript at Stage 1 have now assessed the Stage 2 submission. Both reviews are positive overall, with the Stage 2 criteria largely met. There are, however, some constructive comments to attend to in revision. Reviewer 1 makes a useful suggestion regarding the Discussion and ensuring that the conclusions are appropriate given the evidence. Reviewer 3 makes an interesting point about the definition of anticipatory looks and implies a possible reanalysis using different parameters. This analysis is not necessary to achieve Stage 2 acceptance, as these design characteristics were assessed and approved at Stage 1. However, the authors are welcome to perform additional (transparently exploratory) analyses using different criteria, and should at a minimum respond to this point in the response to reviewers (and possibly the Discussion). Reviewer 3 also suggests some restructuring to present the results immediately after each experiment. This seems sensible provided it does not lead to any unnecessary changes to the approved Stage 1 component of the manuscript, but again is not required and I will leave this to up to the judgment of the authors. Please do switch future tense to past tense as the reviewer requests.

Comments to Author:

Reviewer: 1

Comments to the Author(s)

I have read over this Stage 2 registered report and am satisfied that it meets the standards set out in the Stage 1 report.

I did not see code or materials in the linked OSF repository, only lab logs and preregistration documents. There was also no Dryad link. I thus could not verify that these files were shared.

As the authors write in their lab log, "Experimental tasks 2 and 4 were too difficult for the infants and need to be completely redesigned for this age group." This is a real risk of the RR format - the experiments have limited evidential value for reasons that were difficult for both the authors and the reviewers to foresee. Despite this, I found the authors' discussion of the issues generally appropriate.

I would recommend some minor revision of the discussion section. First, the initial paragraph should be broken up into several distinct paragraphs with different points - at least a clear summary of the current finding and a discussion with respect to related work. Second, the oblique "data from one lab" comment felt odd here because the finding that the authors fail to replicate is from Kovacs & Mehler, not (what I assume to be) the Bialystok group.

Minor:

- "This demonstrates the importance of registered reports" - I don't think this line is warranted in the conclusion. RRs do not guarantee comparable procedures or analyses across groups.

Reviewer: 3

Comments to the Author(s)

1. Are the data able to test the authors' proposed hypotheses by passing the approved outcome-neutral criteria (such as absence of floor and ceiling effects or success of positive controls)? The data analysis follows what had been proposed. In addition, some useful explorative analyses were conducted. Those make a lot of sense and are doing justice to the data.

Having said this, I have a comment on the analyses of Experiments 1 and 2. I noticed that anticipatory looks are defined as saccades within 1s starting 150ms after cue offset. It seems that this is what Kovac and Mehler (2009) did. But I find the start of this period quite restricted. For instance, in Experiment 1, the last cue shape is presented for 0.8s. Isn't it possible that infants move their eyes before that last cue shape disappears? In other words, would an earlier start of that period make a difference to the results? I would have thought 150ms after the onset (not offset) of the last cue would be more appropriate.

2. Are the introduction, rationale and stated hypotheses the same as the approved Stage 1 submission?

Yes, these are the same.

3. Did the authors adhere precisely to the registered experimental procedures?

Yes, they generally did, apart from some very small changes that the authors have marked in red. But these changes do not change the hypotheses or the conclusions.

To note is that procedure and analysis sections are written in future tense. I am not familiar with the policy of the journal, but I would suggest to change them to past tense.

I also would find it helpful if the results of each of the four experiments could follow directly the method section. It is otherwise difficult to remember the details of the experiments.

I would also move the comparison of the socioeconomic status and age of the two participant groups to the participant section.

4. Where applicable, are any unregistered exploratory statistical analyses justified, methodologically sound, and informative?

The additional analyses are very useful and appear sound to me.

5. Whether the authors' conclusions are justified given the data

I generally agree with the conclusions. But I have some further questions and suggestions.

First, it is not entirely clear how all the results are interpreted. For that it would be useful to start the discussion section with a summary of the findings and then make sure that all findings are discussed.

Results of experiment 4 (p. 21):

I don't think that the results of the analysis of the proportion of looks to the changing stimuli necessarily mean that the task was too difficult. It could be that infants notice the change, but loose interest. It would therefore be useful to see a graph for this analysis. (see also comment below about graphs for non-significant results). Btw, I do not understand the comment 'so we cannot conclude that the difference is real' (line 53). Which difference do the authors refer to? The difference of length of looks at the familiar objects? I don't think this analysis diminishes this outcome.

Relation of findings to similar studies:

While there is a comparison of the Kovacs and Mehler (2009) study and other studies (including the present one), I find this unsatisfying. What exactly is different between the current study and the one(s) that find different results? And could one analyse the current study in a way more closely to what these other studies did? For instance, is the method of the current study exactly the same as that in Kovacs and Mehler (2009), apart from the sample size and the 'engaging stimuli'? (Btw, does the latter refer to the cue or the reward stimulus). Also, is the coding exactly the same?

Minor comments:

- Please indicate in the figure captions which experiment they belong to.

- Results of non-significant findings (e.g. Experiment 2): a graph showing the results would be very helpful for the reader.

- P. 10, line 29: change "periphe ral" to "peripheral"

- First paragraph of discussion (p. 22) is too long. Split up into smaller meaningful units.

Author's Response to Decision Letter for (RSOS-180191.R1)

See Appendix D.

Decision letter (RSOS-180191.R2)

05-Feb-2020

Dear Dr D'Souza:

It is a pleasure to accept your Stage 2 Registered Report entitled "Is mere exposure enough? The effects of bilingual environments on infant cognitive development" in its current form for publication in Royal Society Open Science.

Please note that the email address 'd.brady@bbk.ac.uk' is currently unable to receive emails in relation to this manuscript - as we require all authors to have an active email address, please can you confirm with Dr Brady an alternative email address that we may note?

on behalf of Professor Chris Chambers (Subject Editor)
openscience@royalsociety.org

Dear Professor Chris Chambers,

We are delighted to hear Reviewer 1's "overall assessment is that this is a thorough and systematic set of studies that [they] would be excited to see carried out". We are also pleased to hear that "the manuscript is very well written overall and the ideas are very sound" (according to Reviewer 2) and that "the study is very timely" and the "introduction, logic and rationale seem good" (Reviewer 3).

Also, in our experience, many researchers in our field would like to see a replication of the Kovacs and Mehler experiment. So, we wholeheartedly agree with Reviewers 1 and 3 that the replication (Experiment 1) would itself be a useful contribution to the field and have strong implications.

As pointed out, the reviewers' main concerns seem over the possibility of a low proportion of anticipatory looks (Reviewer 1) and the rationale for Experiments 2 (Reviewer 2) and 4 (Reviewers 2 and 3). We have addressed each of the reviewers' comments below (their comments are in green; our response is in black). For example, to satisfy Stage 1 Criterion 6, we now include pilot data and quality checks such as comparing proportion of correct anticipatory looks against chance and including a minimum level of anticipatory looks.

Also, the reviewers suggest we use a more sensitive measure of language background (which we will now do) and Reviewer 1 highly recommends we use a mixed effects analysis (which we will now do).

We thank the reviewers for their helpful comments and suggestions.

Comments to Author:

Reviewer: 1

Comments to the Author(s)

This Registered Report protocol presents an ambitious attempt to extend the state of the art with respect to bilingualism and cognitive control in infants prior to the onset of productive language. My overall assessment is that this is a thorough and systematic set of studies that I would be excited to see carried out, provided my comments are addressed. Even the first experiment (the Kovacs & Mehler 2009 replication) would itself be quite a useful contribution. Despite my enthusiasm I had some specific concerns regarding the tasks, theoretical grounding, and proposed statistical analyses, detailed below, that would make me worried about going forward without seeing revisions to the protocol.

First, anticipatory looking paradigms are quite tricky. Without extensive piloting/paradigm timing customization, it can be difficult to see any anticipation at all. Many labs that use this kind of paradigm report that extensive piloting is necessary to ensure an adequate level of anticipation (without which there is no way to see differences on the manipulation of interest). It would be quite disappointing to see the results of Experiments 1 and 2 be uninterpretable due to a low proportion anticipations. I am not sure how to address this as E1 in particular is a replication study and some aspects of a failure might be due to issues in the original experiment. At a minimum, some discussion of failures to anticipate and some analyses of the proportion of anticipations seems critical for interpretability. More generally, some presentation or discussion of the age-appropriateness/validity of the paradigms being used would be useful. If the authors have pilot data it would not be inappropriate to show them in the RR to demonstrate that basic methodological issues (retention, data loss) have been piloted. (In a grant situation, I would vastly prefer such pilot data because they would signal the feasibility of measuring the effect of interest).

We agree it would be a shame if the results of Experiments 1 and 2 are uninterpretable due to a low proportion of anticipatory looks. For Experiment 1 (the replication), we believe we are using more

interesting stimuli than Kovacs and Mehler (2009); and of course, we are using the same timings as them and yet a much larger number of participants (51 vs. 20 per group). To maximise the number of children engaged in the study, we are also using more rewards than Kovacs and Mehler (2009).

To interpret our data (and satisfy Stage 1 Criterion 6), for each group (monolingual, bilingual) we will now test whether correct anticipatory looks increase over the *trials* (pre-switch, post-switch) and, if they do, whether the infants made significantly more correct anticipatory looks than expected by chance (pp. 6-7). Furthermore, we would only conclude that a group of infants can anticipate the side a reward will appear on when at least half of them have contributed anticipatory looks (p. 7). Therefore, not only will we directly compare bilinguals to monolinguals, not only will we try to replicate Kovacs and Mehler's (2009) finding that monolinguals fail to switch, but we will check that the monolinguals made increasingly (and significantly) more correct anticipatory looks *pre-switch*. If we find that they did not, then we will conclude that their performance was at floor (and uninterpretable). In other words, if infants make increasingly more correct anticipatory looks *pre-switch* that become significantly greater than chance level, then we will be well placed to interpret the post-switch data.

We now include pilot data for Experiment 1 (see Supplementary Information below). Collapsing across groups ($N = 14$), the Experiment 1 pilot data show that for trials 7-9 (pre-switch), proportion of correct anticipatory looks—i.e., $\text{correct}/(\text{correct}+\text{incorrect})$ —averaged over the last three pre-switch trials (.81) was significantly greater than chance (.50), $t = 3.00$, $df = 13$, $p = .010$ (2-tailed). This suggests that infants can learn to anticipate a reward in this experimental task.

Moreover, when we use the same measure as Kovacs and Mehler (2009)—i.e., $\text{correct}/(\text{correct}+\text{incorrect}+\text{no anticipatory look})$ —then our measure is like those reported in Kovacs and Mehler (2009). For trials 7-9 (pre-switch) and 10-12 (post-switch), our proportion of correct anticipatory looks to all looks is .50 and .17; Kovacs and Mehler report $\sim .48$ and $\sim .16$, respectively (see Fig. 4 in Kovacs & Mehler, 2009). For the final three (post-switch) trials (16-18), we find .19 with a greater proportion of *monolingual* to bilingual (4:3) infants; Kovacs and Mehler report $\sim .25$ (1:1).

The Experiment 1 pilot study is identical to Experiment 1 proposed in this study. We are therefore confident that a low proportion of anticipatory looks will not be an issue for Experiment 1.

We also now include pilot data for Experiment 2 (see Supplementary Information at the end of this document). The Experiment 2 pilot study was similar—but more challenging—than Experiment 2 proposed in this study. The Experiment 2 pilot study contained fewer trials (by half) and more degraded (fragmented) figures than the ones proposed in this study. Nevertheless, proportion of correct anticipatory looks (.63)—i.e., $\text{correct}/(\text{correct}+\text{incorrect})$ —was trending above chance level (.50), $t = 2.01$, $df = 13$, $p = .065$ (two-tailed). Importantly, over the final four familiarization trials, proportion of correct anticipatory looks (.72) was significantly greater than chance (.50), $t = 3.12$, $df = 11$, $p = .009$ (two-tailed). This suggests that the infants can learn to anticipate a reward in this experimental task.

Although for the test trial a binomial test indicated that proportion of correct anticipatory looks (.75) was not significantly higher than the expected .50 ($p = .289$), proportion of correct anticipatory looks was—as hypothesised—higher, albeit *not* significantly higher, in bilinguals (.80) than monolinguals (.67) (see Supplementary Information).

For the test trial, six infants failed to saccade within the anticipatory period. However, the data give us confidence that proportion of correct anticipatory looks will not be at floor in the *less challenging* version (more trials, less degraded pictures) we would like to carry out.

(For completeness, we have now added Experiment 4 pilot data to Supplementary Information below. For all three experiments, you can access the raw and processed pilot data using the following link: <http://datadryad.org/review?doi=doi:10.5061/dryad.6sn36>. We have no need to pilot Experiment 3, because the gap-overlap task is a well-established paradigm and the specific version we will be using has been developed and piloted over many years at the Centre for Brain and Cognitive Development, Birkbeck, University of London; see D'Souza, 2014, for details.)

To satisfy Stage 1 Criterion 6, we have also added a minimum level of anticipatory looks (50%) to training and test phases in Experiment 2 (pp. 8-9).

Theoretically, I found that the introduction did not seem to take seriously the possibility that bilingualism findings with preverbal infants might be artifactual or incorrect. There are at least four theoretical possibilities in this space (logically):

1. bilingual CC advantage caused by production (attributed to Green),
2. " caused by home environmental (attributed to D'Souza & D'Souza),
3. bilingual CC advantage due to other causes, and
4. No bilingual CC advantage

The proposed experiments are informative regarding this space, with failures to observe differences consistent with both 1 and 4 and successes consistent with 2 or 3. I thought that a little more care in navigating this theoretical space was warranted.

Taking seriously the possibility that bilingualism findings might be artifactual or incorrect

The most influential explanation for the purported bilingual advantage cannot explain why preverbal infants show a bilingual advantage (Kovacs & Mehler, 2009; Singh et al., 2015). Although we have put forward a theory that can explain it, by including a replication of Kovacs and Mehler (2009) we feel we are taking seriously the possibility that bilingualism findings with preverbal infants might be artefactual or incorrect. We believe that many scientists would like to know whether the study can be replicated or not. In fact, one of the authors is deeply sceptical about the bilingual advantage, one of us is not, and two of us are uncommitted to the thesis – so, we are also curious to learn whether Kovacs and Mehler (2009) can be replicated with a larger sample.

However, to ensure it is clear we take the possibility seriously, in our opening paragraph we stated that "inconsistent results" and "publication bias" have "led many scientists (e.g., Duñabeitia & Carreiras, 2015; Klein, 2016; Paap, Johnson, & Sawi, 2016) to question whether the bilingual advantage is real or merely an artefact of particular research practices" (p. 1). We have now added the following sentence to the opening paragraph: "Moreover, significant bilingual advantages have not been observed in studies with large sample sizes (see Paap, Johnson, & Sawi, 2016, for discussion)" (p. 1). Also, we have now replaced the *word* show with *suggest*: "some studies suggest there is a bilingual advantage in preverbal infants" (p. 2).

Also, we now end the introduction with the following sentence: "If we fail to observe a significant group difference, we will argue that there is insufficient evidence to suggest there is a bilingual advantage in preverbal infants" (p. 3). Therefore, if we fail to observe a difference, when we relate our findings to the literature in the Discussion section, we will discuss the possibility that there is not enough evidence for anyone to conclude that there is a bilingual advantage in preverbal infants.

The four theoretical possibilities

Although we agree that there are four theoretical possibilities, we are not attempting to differentiate between 2 and 3. We are arguing "that exposure to more varied language environments drive infants to explore (sample) further by constructing less detailed models of their environments and placing more weight on novel information" (p. 3). Thus, our aim is to compare two groups that differ only on language exposure ("monolinguals" vs. "bilinguals"). Any group difference will be attributable to

differences in language *exposure* (possibility 2-3) and not language *production* (possibility 1). We will now use a more sensitive measure of language exposure – see below.

However, we cannot disprove 1. That is, we cannot use a group of preverbal infants to disprove that at least one of the underlying mechanisms in older children and adults involves language production. Therefore, to ensure clarity, we have added the following sentences to the end of the introduction: “Although these infant studies can neither support nor disprove the theory that a bilingual advantage results from managing two or more languages during production in older children and adults, if we observe a significant group difference we will argue that it reflects experience-dependent adaptations that occur because of regular *exposure* to two or more languages. If we fail to observe a significant group difference, we will argue that there is currently insufficient evidence to conclude there is a bilingual advantage in preverbal infants” (p. 3).

In other words, we are arguing that 1 does not explain all the available data (e.g., Kovacs & Mehler, 2009). We therefore want to see whether we can replicate Kovacs and Mehler (2009) and, if we can, suggest and test the hypothesis that language exposure leads to adaptive differences in the trade-off between exploration and exploitation.

On the topic of participants: Why not use a standardized instrument for gathering language background? Self report of percentages is problematic and many folks have worked on this extensively. There are several well-validated instruments, e.g.:

- Cattani, A., Abbot-Smith, K., Farag, R., Krott, A., Arreckx, F., Dennis, I., & Floccia, C. (2014). How much exposure to English is necessary for a bilingual toddler to perform like a monolingual peer in language tests? *International Journal of Language and Communication Disorders*.
- Li, P., Zhang, F., Tsai, E., & Puls, B. (2014). Language history questionnaire (LHQ 2.0): A new dynamic web-based research tool. *Bilingualism: Language and Cognition*.

Likely the background of the infants tested will be one of the major interpretive points in parsing the results - it is critical then to have a strong characterization of language background. The current short questionnaire is subject to biases in reporting (self report of percentages is quite noisy) and quite brief.

We thank the reviewer for pointing this out and have decided to use Cattani et al.’s (2014) Language Exposure Questionnaire (as also suggested by Reviewer 3).

Other comments, mostly on analysis:

* Will the authors confirm with parents that children are not yet producing words? This will be true for the majority of children, but surprisingly, there are some children in the 8-9 month range that have produced a first word.

We will confirm this with parents by asking them when they come to the lab and including the Oxford CDI (<https://www.psy.ox.ac.uk/research/oxford-babylab/research-overview/oxford-cdi>) for each language their child is exposed to. The OCDI is a list of 416 words that are commonly acquired in infancy. For each word, the caregiver indicates whether their child can “understand” and/or “understand and say” the word. We will use the OCDI to confirm that their infant is not yet producing words. If we find any group differences, then we may explore the effect in bilinguals using the receptive language measure (“number of words understood”). We would expect that the more words a bilingual child understands from two or more languages, the more they are “embedded” in bilingualism. It would be interesting to use it as a continuous measure of “bilingualism” and find out whether it covaries with the observed effect. We would do this in a separate “exploratory” section.

We have now included this in the manuscript (p. 12).

* K&M2009 analysis: "To test this hypothesis, we will measure proportion of correct anticipatory looks averaged over the final three trials of the post-switch phase. The proportional data will be arcsine transformed and analysed using an independent samples t-test." A more sensitive analysis would be logistic mixed effects regression, which could account for trial order and would not require a transform. I'd strongly recommend that the authors consider that model as the primary analysis (unless they want to designate a "replication analysis" and preregister that too - the original authors appear to have used an f-test with no transform).

We thank the reviewer for their recommendation. We will follow the reviewer's recommendation and carry out a logistic mixed effects regression (using the glmer function of the R package lme4; Bates, Maechler, Bolker, & Walker, 2013). That is, the unit of observation will be *one trial per child*. The outcome variable will be *anticipatory look* (correct vs. incorrect). The predictor variables will be *trial* (1-9) and *group* (monolingual vs. bilingual). Random coefficients and an autoregressive covariance structure will be assumed. (See pp. 6-7.)

Reference

Bates, D., Maechler, M., Bolker, B., & Walker, S. (2013). lme4: Linear mixed-effects models using Eigen and S4. R package version 1.0-5. <http://CRAN.R-project.org/package=lme4>

* Do the authors want to take further advantage of trial order and model the trend over more than 3 trials (increasing power), as in the linear trends shown in K&M2009's original paper?

We will follow the reviewer's recommendation and model the trend over 9 trials (two models: pre-switch, post-switch) (pp. 6-7).

* "We will run independent samples t-tests to check that the two groups did not significantly differ from each another on age, gender, and parents' SES." Another, potentially more powerful, possibility would be to control for these in planned analyses, e.g. the model described above.

Although we could control for age, gender, and parents' SES, we would rather select a small set of predictors based on theory. And because (like Kovacs & Mehler, 2009) we will be selecting participants based on their age, gender, and parents' SES, we do not expect that these factors will vary much in the final sample. The groups will be the same age and come from similar socioeconomic backgrounds. We have no reason to believe that gender will contribute to performance, though we will check that the (roughly 1:1) ratio does not significantly differ across groups. We think the paper would be simpler (and thus more accessible to the reader) if we could treat the two groups as two groups that differ only on language. We would therefore like to recruit two well-matched groups and start the analyses by testing our assumption using *t*-tests. If there are no significant differences between the groups on age, gender, and/or SES, then we would proceed with the assumption that the two groups differ only on language. However, if we find large variation and/or significant differences between the groups, then we could explore it using ad hoc analyses in a clearly designated "exploratory" section.

* "For each experimental task, we will report any statistically significant difference between groups on the number of trials that data were provided." Wasn't sure what this meant.

We have now clarified this: "For each experimental task, we will run independent *t*-tests to check that the two groups did not differ on the number of valid trials that they provided" (p. 4). If one group produces significantly more invalid trials than the other group, it could affect the interpretation of our results.

Reviewer: 2

Comments to the Author(s)

This paper proposes to carry out a series of eye-tracking experiments with a cohort of 102 7- to 9-month-olds (half bilinguals, half monolinguals), to revisit the claims that bilinguals develop a cognitive advantage.

The manuscript is very well written overall and the ideas are very sound. I am particularly eager to see a large-scale replication of Kovacs and Mehler (2009). Indeed the demonstration that 7-month-old bilinguals had better inhibitory skills than monolinguals has strong implications, and as such, a replication is very welcome.

I am also very interested in the extension of the investigation to other aspects of cognition such as attention shifting (Exp 3), but a bit more sceptical about the claims that bilingual children would be sampling less visual information (Exp 2 and 4). I am not convinced by the logic according which more complex environments would lead to impoverished representations of the world. For example, in the acquisition of morphosyntax, kids who are acquiring more complex morphologies are not delayed as compared to those who are learning simpler ones, they do reach an equal level of expertise within the same timeframe. However I would be very interested to see the outcomes of exp 2 and 4.

One caveat is the questionnaire used to calculate exposure: asking parents their evaluation of the amount of time each member of the family speaks language A or B does not sound like the most accurate way to estimate exposure – and exposure is a key factor here as it helps determine whether a child is in group A or B. There are a few available questionnaires out there which are more accurate (and not long at all, see for example the Plymouth LEQ which is online at <http://www.psy.plymouth.ac.uk/babylab/languageexposure.html>)

We thank the reviewer for their positive comments.

Although we argue that experience-driven adaptations are a function of complexity in the environment, we are not suggesting that these adaptations will result in developmental delay. On the contrary, we expect the adaptations to result in a similar level of expertise in everyday language use within the same timeframe – albeit it with “side effects” such as bilinguals outperforming monolinguals on tasks of cognitive control, and monolinguals outperforming bilinguals on tasks that measure memory of fine detail. This has not hitherto been tested, so we would be very interested to see the outcome too.

We thank the reviewer for their suggestion to use the Plymouth LEQ. We have now added the LEQ to our protocol (see pp. 4 & 12).

Reviewer: 3

Comments to the Author(s)

This is in general a very interesting study that will test an alternative account of bilingual cognitive advantage in infants. There is currently a hot debate about the bilingual cognitive advantage, and the infant data are puzzling. Thus, the study is very timely.

The introduction, logic and rationale seem good. I do not have any comments or questions about them. I do, however, have some questions about the materials and methods. Some of these are just clarification questions, others more critical. Importantly, I am not convinced (yet) that Experiment 4 tests what it is meant to do.

We thank the reviewer for their positive comments and address their concern about Experiment 4 below.

p. 4, line 20. Participants: Please state how you will measure parents' socioeconomic status.

We will use our background questionnaire (Supplementary Information) to measure parents' socioeconomic status (SES). Specifically, our SES measure will comprise an index of socioeconomic deprivation (the primary caregiver's *postcode*) and the caregivers' *educational attainment, income, and occupation(s)* – see details below.

We thank Reviewer 3 for spotting this omission.

Postcode

As an index of socioeconomic deprivation, an individual's postcode is highly predictive of cognitive ability (Nettle, 2010; Pepper & Nettle, 2017). For each participant, we will use their postcode and UK government data to obtain a "deprivation" rank score (<http://dclgapps.communities.gov.uk/imd/idmap.html>). We will rescale the ranks to make them easier to interpret, using the following formula:

Postcode score = deprivation rank / the number of neighbourhoods in England

So, the postcode of the median England neighbourhood would have an SES score of 0.5 (i.e., 16241/32482 = 0.5).

Education attainment, income, and occupations

We will also obtain a mean score for caregivers' education attainment, income, and occupations.

Education attainment (1-7) and household income (1-11) scores will be obtained from the background questionnaire and rescaled to fall between .00 and 1.00. For example, an education attainment raw score of 5 will become .71 ($5/7 = .71$).

Occupations will be obtained from the background questionnaire, and scored using "prestige" scores reported in the National Opinion Research Center's *Colorado Adoption Project: Resources for Researchers* (<http://ibgwww.colorado.edu/~agross/NNSD/prestige%20scores.html>). This lists 888 occupations, from Physician (prestige score: 86.05%) to 'Miscellaneous Food Preparation Occupations' (prestige: 16.78%). Percentages will be converted to proportions (e.g., 86.05% will be converted to .86).

If the infant has two parents, then the education and occupation scores will reflect the mean average score of both parents.

SES score

The final SES score will be the mean average of all four weighted scores:

SES score = ((postcode x 3) + education + income + occupation) / 6

The SES score can therefore range from .067 to .977 (the score cannot go lower than .067 because the lowest education, income, and occupations scores are .143, .091, and .168, respectively; nor higher than .977, because the highest occupation score is .861).

Because of the postcode's predictive power (Pepper & Nettle, 2017), we have decided to give postcode equal weight to a combined score of education attainment, household income, and occupation.

We have now added this information to page 4 and Supplementary Information.

References

Nettle, D. (2010). Dying young and living fast: variation in life history across English neighborhoods. *Behavioral Ecology*, *21*(2), 387-395.

Pepper, G. V., & Nettle, D. (2017). The behavioural constellation of deprivation: causes and consequences. *Behavioral and Brain Sciences*, *40*, e314.

p.4, line 24. It is stated that results of a particular infant are included into the analysis of a particular experiment when the infant has at least 75% usable eye tracking data for that experiment. But what will you do if an infant produces 75% good data overall, but only 1 or 2 valid trials out of the small number of trials you actually analyse (these are never more than 3 trials). It seems to me that these critical trials need to be valid trials as well.

We agree that infants must provide data for critical trials as well as provide at least 75% useable eye tracking data.

On the advice of Reviewer 1, we will now be running mixed-effects logistic analyses, which avoids the problem that arises from using *t*-tests to analyse a small number of trials. However, for Experiment 2 we are mainly interested in looking behaviour during two critical test trials, so we are still faced with the problem that some infants may provide 75% useable eye tracking data overall but no eye tracking data for the two test trials. Therefore, we have added to the manuscript the following sentence: "Infants must provide useable data for at least one of the two test trials in Experiment 2" (p. 6).

We thank the reviewer for spotting this omission.

Although we hope that most infants will provide useable data for both test trials, infant habituation studies suggest that one test trial is often sufficient.

p.4, line 29: It is stated that if not all infants produce enough valid trials for all experiments, that more infants are tested so that there will be 51 infants of each participants group in all experiments. This will mean that some infants won't take part in all experiments. Nevertheless, it seems to me useful to analyse whether performance on the 4 experiments is correlated. In other words, if a particular infant shows superior performance in one experiment, is he likely to do this also on the other experiments? Otherwise it might be that different bilingual infants might drive a bilingual superiority effect in different experiments. If so, one cannot generalise to bilingual infants as a population.

We agree it would be theoretically interesting to analyse whether performance across tasks is correlated. We could add this analysis to an exploratory section at the end of the Results section. Because it would involve looking at individual differences, we will now present the experimental tasks in fixed order (Mollon, Bosten, Peterzell, & Webster, 2017).

However, for practical reasons, we cannot promise to keep testing until we have good data across all four experiments from the same 102 infants – but we hope to obtain a large enough sample to explore this interesting question. To detect an expected large-sized effect (0.5), we would need a total sample size of 26 (power = .80, α = .05 (two-tails); calculated using G*Power).

Reference

Mollon, J. D., Bosten, J. M., Peterzell, D. H., & Webster, M. A. (2017). Individual differences in visual science: What can be learned and what is good experimental practice?. *Vision Research*, *141*, 4-15.

p. 4, line 42: what is the purpose of the camera that is mounted directly above the horizontal midpoint of the screen? It is stated that this will monitor and record infant behaviour. Why is this important? These recordings do not seem to be analysed or used in any way.

We will use the camera to monitor parent/infant behaviour while we sit behind a screen. If an infant becomes irritable, we can abandon the experimental task. Because these experiments are so short, we do not anticipate abandoning many experimental tasks. We will save the recordings in case we would like to later check the child's behaviour (e.g., if the eye tracking data look strange). Recording the baby might also be useful for the co-authors who are not collecting the data – i.e., so they can check what the baby and researchers are doing. However, in practice, we usually delete video recordings at the end of each study to save storage space.

p. 6, Experiment 1, coding and analysis:

It is stated that if an infant looks at both the left and right side of the screen during the critical period, that the longer look will be coded. Do you mean that if an infant looks 51% of the time to the left side and 49% of the time to the right side, that you will treat their look as if they were looking only to the left side? That does not seem to be right as it does not account for the actual looking behaviour. I would suggest to use a stricter criterion. For instance, that the longer look has to be at least 75% of the looking behaviour.

Because the anticipatory time window is only 1 s long and precedes the reward stimuli, few infants will shift their attention to both sides. In Kovacs and Mehler's study, the infants looked to both sides in less than 5% of the trials, and "taking the first look yielded practically identical data" (Kovacs & Mehler, 2009, p. 6559).

Related to that, why are looks coded as right, middle and left, and not whether the infants look to the white boxes where the rewards will appear? (see also other experiments) Left and right seems a very rough measure if you have an eye tracker that provides more precise information.

We will now only analyse looks that fall within two predefined Areas-Of-Interest (AOI). The predefined AOIs will extend 1 cm around the white boxes in case the infants look at the outer edge of the reward stimuli (which touch the edges of the white boxes) and/or the calibration is slightly off.

We have added this to the manuscript (p. 6).

p. 6, line 38: In order to be sure that infants have actually learned to anticipate the reward in the pre-switch phase, you will need to also analyse the proportion of correct anticipatory looks averaged, for instance, over the final three of the pre-switch phase.

On Reviewer 1's suggestion, we will now analyse all trials, including the final three trials of the pre-switch phase. We will analyse all 9 pre-switch trials, using a logistic mixed effects regression. Our aim is to replicate Kovacs and Mehler's findings and show that number of correct anticipatory looks significantly increases over the 9 pre-switch trials (in both groups – and with no significant difference between the groups).

p.6 line 38: Related to my question above: What if not all three final post-switch trials are valid trials? Will you take the 1 or 2 valid trials instead? That seems a very small number of trials to draw a strong conclusion from.

We will now be analysing all 9 post-switch trials, using a logistic mixed effects regression. We think it would suffice to replicate Kovacs and Mehler's findings and show that number of correct anticipatory

looks significantly increases over the 9 post-switch trials in the bilingual group but not in the monolingual group.

p. 6, line 41: Here and for at least one other experiment, it is stated that infants looking behaviour will be compared against chance. Please state what exactly is a chance behaviour. Do you mean by chance a proportion of 0.5 to the correct side? I would find this problematic. Looking left or right are not the only options that the infant has. They can also look to the middle of the screen or not at all to the screen.

We will now only analyse looks that fall within two predefined Areas-Of-Interest (AOI). If infants look significantly more at one AOI than the other AOI, we will conclude that they did not randomly allocate their attention. To account for any side bias, we will counterbalance the order of stimulus presentation, for all experiments.

p. 7, Experiment 2, figure 2:

Can you please also add the figures of the elephant that you will use as stimuli.

We have now added this (p. 7).

p. 8, line 14: two trials seem very little to draw conclusions from (especially if one of them is not a valid trial). Are two trials enough?

Although two trials are not ideal (some infants will look away during the critical trials), evidence in the infant habituation literature suggest that one or two trials are usually sufficient to detect an effect at the group level.

Experiment 3

p. 9: where does the reward appear?

The reward appears in the same place as the target. We have now added this in (p. 9) and thank the reviewer for spotting this omission.

p. 9, line 18: how do you code a trial where the infant shifts their gaze away from the central fixation stimulus, but not towards the target?

If the child shifts away from the central fixation stimulus but does not gaze towards the target, then the trial will automatically be coded as invalid. We have now included this in the manuscript (p. 10).

p. 9, line 23: why does one need to subtract RTs in the baseline condition from RTs in the overlap condition? Why can one not simply use the RT in the overlap condition as a measure of failure to disengage?

We will not be analysing "failure to disengage" (which occurs when a child fails to shift their attention from the central fixation stimulus). This is because we fully expect 7- and 8-month-old infants to disengage attention in our paradigm and fixate on a visual target within 1.2 s (this assumption is based on previous studies using the same paradigm; D'Souza, 2014; D'Souza, D'Souza, & Karmiloff-Smith, 2017). So, there will be few (if any) true failures to disengage.

However, there will be individual differences in how quickly infants can shift attention in the overlap (competing stimuli) condition. Therefore, it's important to subtract RTs in the baseline condition from RTs in the overlap condition, to account for any individual differences in baseline reaction time. This will yield a score that will reflect the cost of disengaging attention. In other words, the only difference

between the baseline and overlap conditions is the presence of a central fixation stimulus in the overlap condition. So, any difference between the baseline and overlap conditions must be attributable not to differences in *shifting* visual attention but to the ability to *disengage* attention from a visual stimulus. A larger disengagement effect (overlap RT minus baseline RT) will mean that the infant is slower at disengaging attention from a visual stimulus.

P. 9, line 25: I do not understand why the gap condition is a control for fast saccading. Please explain.

The gap condition includes time (200 ms) for the infant to fully disengage their attention from a central fixation stimulus prior to the appearance of a target stimulus, so it can be seen as a pure measure of how quickly the infant can *shift* their visual attention. Infants respond to the target stimulus faster in the gap condition than the baseline condition (e.g., D'Souza, 2014; D'Souza, D'Souza, & Karmiloff-Smith, 2017). We do not expect to see any difference in the "pure" ability to shift attention to a visual target. We just want to run the analysis as a check that there are no group differences in visual-motor skills.

Experiment 4:

-P. 9, line 38: state here what the line drawings show. It would be useful to have some examples of how the line drawings look when changing over the 15 trials, especially of the critical trials that are analysed. Do the critical trials actually show meaningful objects?

We now include examples of the line drawings (p. 10). The line drawings change from a human male face to a woman holding flowers. Although the line drawings would be meaningful to an adult, they may not be meaningful to a child. But we hope they are sensitive to the change.

We will now be analysing all 15 trials, so there are no critical trials now.

p. 9, line 52: what is the purpose of the background music?

Unlike the other experiments, Experiment 4 does not contain any rewards – so we have added background music to ensure that there is no silence and to create a "warmer" environment.

p. 10, line 3: it is stated that the number of switches are measured. Is this on trials 2 and 3, or all 15 trials?

Initially, this was on trials 2 and 3. However, we will now be analysing data from every trial using a mixed-effects logistic regression (see our reply to Reviewer 1 and p. 11), so we will now be measuring number of switches for all 15 trials.

p. 10, line 4: please define 'familiar' stimulus and 'novel stimulus/side'. I don't understand.

On the first trial, the same stimuli (S1) will be presented on both sides simultaneously. From the second trial onwards, however, the child will see S1 on one side of the screen and a novel stimulus (S2) on the other side of the screen. So, in the second trial the child will see S1 (a familiar stimulus) on one side of the screen and S2 (a novel, albeit only slightly different) stimulus on the other side of the screen. In the third trial, the child will see S1 (a familiar stimulus) on one side of the screen and S3 (a novel, albeit only slightly different) stimulus on the other side of the screen. In every subsequent trial, the infant will see a familiar stimulus (S1) on one side of the screen and a novel stimulus (S4, etc.) on the other side of the screen. The novel stimulus will always appear on the same side (the "novel side").

p. 10, line 12: why do you look at trials 6 and 7?

We believe that the middle trials (6 and 7) are when the infant is most likely to notice a change. However, we will now be analysing data from every trial using a logistic mixed effects regression (see our reply to Reviewer 1 and p. 11).

p. 10, line 4/5. It is stated that bilinguals are expected to switch more frequently because they err on the side of exploration compared to monolinguals. But is it not also possible that frequent switches mean that an infant realises the change and is interested in what is going on at both sides? Checking whether both sides are changing or only one side? And why would monolinguals rather look at the familiar (=non changing picture) when they want to build detailed models? And how can you tell that both groups of infants can equally discriminate between the novel and familiar picture? And why is looking at the changing picture evidence of not remembering the change? If bilinguals explore a lot, they will notice the change at trials 6 and 7 because these will look very different from what they had seen at the beginning of the experiment. In sum, I am afraid I am not convinced that this experiment actually measures what it is supposed to measure.

Experiment 4 yields two measures: one that indicates whether the infant looked significantly longer at one stimulus than the other (i.e., whether the infant noticed a change), and one that indicates how the infant explored the visual stimuli. If the infants look significantly longer at one stimulus than the other, then we can conclude that they are discriminating between the two (to account for any side bias, side of presentation is counterbalanced). If the infants fail, from trial 1 to trial 15, to significantly look longer at one stimulus than the other, then we can draw no firm conclusions.

If one group of infants switch between stimuli significantly more than the other group of infants, then we can conclude that one of the groups was exploring both stimuli more than the other group.

We do not think that frequent switches reflect awareness of a change. For example, when we tested the paradigm on adults, the moment the adult noticed the change, he or she made fewer switches and focused more on the novel stimulus. So, number of switches (a proxy for 'exploration') will not tell us whether the infant realised that there is a change. We hope that it will tell us whether the bilingual group is more likely than the monolingual group to explore the features of both stimuli rather than concentrate their efforts on one stimulus at a time.

We argue that to build up detailed representations, one must focus on one stimulus of interest at a time (Mareschal et al., 2007). If an infant frequently switches between stimuli, then he or she will spend less time focusing on the internal features of a single stimulus and thus less time building up a detailed model of any one stimulus. We expect *both groups* to switch between stimuli, but we hypothesise that bilinguals will switch significantly more than monolinguals; we expect that monolinguals will focus more on the internal features of a stimulus (rather than spending time switching between the two stimuli) and thus notice the change sooner (albeit with fewer switches between stimuli). Moreover, we expect that even after noticing a change, monolinguals will spend significantly more time *than bilinguals* processing the familiar (vs. novel) stimulus.

Looking for significantly longer at the novel (different) stimulus is indeed evidence that the infant remembers that one of the stimuli is familiar (unchanged). And we agree that if the bilinguals explore a lot, then they may notice the change at trials 6 and 7 – our argument is that monolinguals will be building up more *detailed* representations of the stimuli and thus are more likely to notice changes. In other words, we hypothesise that switching a lot may be useful in many contexts but are not optimal for Experiment 4. This is because the differences between the stimuli are very slight – so switching more frequently is unlikely to confer much of an advantage *unless* the infant has invested time in building up a very detailed model of the familiar stimulus. We hypothesise that the monolingual infants will switch less and thus spend more time absorbing the details (and thus more likely to notice any changes when switching attention).

We thank the reviewers and editor for their helpful feedback. We also changed the Supplementary Information accordingly, and made a few minor changes to the main document to improve our submission (we tracked changes). We believe that our manuscript has been significantly strengthened.

Yours sincerely,

Dean D'Souza

Supplementary Information – Pilot study

The paradigm we will be using for Experiment 3 (the gap-overlap task) is well established and has been used and developed extensively (by us and other research groups; e.g., D’Souza, 2014). To evaluate the feasibility of Experiments 1, 2, and 4, pilot data were processed ($n = 14$, 8 monolinguals, 6 bilinguals). Because of the small sample size, we did not separate infants into two groups. Please note that some of the infants had taken part in a battery of experimental tasks beforehand (i.e., in a different study for a different research team) and were showing signs of fatigue and fussiness. We included these infants in order to process more data (because our grant does not cover the collection of pilot data).

Experiment 1 pilot data

Overall, infants provided data for 97% of the trials (i.e., for 244 of 252 trials). Nine participants provided data for all 18 trials. All but one participant provided data for at least 75% of the trials in the pre-switch phase and at least 75% in the post-switch phase. One participant failed to provide data for the first three post-switch trials. As one might expect, only 116 of all looks occurred within the narrow time window and were coded as ‘anticipatory looks’. Importantly, all participants provided at least one anticipatory look in the final three pre-switch trials, and nine infants provided anticipatory looks for at least two of the final three pre-switch trials. A one-sample t -test shows that proportion of correct anticipatory looks—i.e., correct/(correct+incorrect)—averaged over the last three pre-switch trials 7-9 (.81) was significantly greater than chance (.50), $t = 3.00$, $df = 13$, $p = .010$ (2-tailed). Our tentative conclusion is that after 6-8 trials, infants can anticipate the side a reward will appear on.

Fig. SI-1. Experiment 1 was piloted on 14 infants. Over trials 7-9, proportion of correct anticipatory looks (.81) is significantly greater than chance (.50). Our tentative conclusion is that after 6-8 trials, infants can anticipate the side a reward will appear on. From trial 10 onwards, the reward appeared on the other side of the screen. Experiment 1 of the main study will seek to replicate this finding and ascertain whether, during trials 10-18, bilingual (but not monolingual) infants can inhibit their learned response to learn a new rule.

Experiment 2 pilot data

The Experiment 2 pilot study was similar—but more challenging—than Experiment 2 proposed in this study. The Experiment 2 pilot study contained fewer trials (by half) and more degraded (fragmented) figures than the ones proposed in this study. That is, to pilot the test stimuli and check

that they were not too fragmented, we presented infants with 4 training trials and 12 test trials (rather than 24 training trials and 2 test trials). Nevertheless, infants provided data for 87% of the trials (i.e., for 194 of 224 trials). Twelve of the 14 infants provided data for at least 75% of the trials. One infant provided data for more than 75% of the trials but looked away for the last two trials.

Only 117 looks occurred within the narrow time window and were coded as ‘anticipatory’ looks. A one-sample t -test shows that proportion of correct anticipatory looks—i.e., correct/(correct+incorrect)—averaged over all trials (.63) was trending but not significantly greater than chance (.50), $t = 2.01$, $df = 13$, $p = .065$ (two-tailed).¹ Importantly, over the final four (of 16) trials, proportion of correct anticipatory looks (.72) was significantly greater than chance (.50), $t = 3.12$, $df = 11$, $p = .009$ (two-tailed). This suggests that infants can learn to anticipate a reward in this experimental task.

Although the infants received few training trials, on presentation of the first test trial, six of the infants anticipated the reward, two made anticipatory looks in the wrong direction, and six failed to saccade within the anticipatory period. Interestingly, proportion of correct anticipatory looks was—as hypothesised—higher, albeit not significantly higher, in bilinguals (.80) than monolinguals (.67) (see Figure SI-2 below). However, we cannot draw firm conclusions from this analysis; a binomial test indicated that the proportion of correct anticipatory looks (.75) was not significantly higher than the expected .50 ($p = .289$).

Fig. SI-2. Although infants received few training trials, on presentation of the first test trial, infants made more correct anticipatory looks than incorrect (*but this result was not significant*, $p = .289$). Furthermore, as hypothesised, more bilinguals correctly anticipated the reward than monolinguals (*but please note that this difference is unlikely to be statistically significant*).

Experiment 4 pilot data

For the critical trials 6 and 7, all 14 infants provided anticipatory looks. As predicted, one sample t -tests show that proportion of looks was significantly greater than chance (.50) in the monolingual group ($t = 2.73$, $df = 7$, $p = .029$, two-tailed) but not in the bilingual group ($t = 0.28$, $df = 5$, $p = .788$, two-tailed). Furthermore, as predicted, bilingual infants switched significantly more than monolingual infants, $t = 2.65$, $df = 12$, $p = .021$ (two-tailed, calculated as average number of switches per individual; see Figure SI-3).

¹ Because this is a small-n pilot study, we included data from the two infants who provided data for less than 75% of the trials. However, in the proposed larger-n study, these two infants would have failed to meet the inclusion criteria and their data would not be analysed.

Fig. SI-3. As hypothesised, over the duration of the experiment, bilingual infants switched between two stimuli significantly more than monolingual infants.

Appendix B

Dear Editors,

We are delighted to hear that our Manuscript RSOS-180045 entitled “Is mere exposure enough? The effects of bilingual environments on infant cognitive development” has been accepted in principle for publication subject to minor revision. Please see below for our response (in black) to Reviewer 3’s comments (in green).

The authors provide a lot of explanations about the processes in Experiment 4 that make a lot of sense and that I fully agree with. But I believe that the logic, the potential results and their interpretation have still not been fully thought through.

I agree that more frequent switching between the two stimuli means more exploration than less frequent switching. And the pilot data nicely suggest that the authors will find that bilingual children generally switch more frequently and therefore seem to be less eager in gaining detailed information about the stimuli than monolinguals.

I also agree that if the monolingual (or bilingual) infants as a group look longer at one of the two stimuli (changing or non-changing) that they seem to notice a difference between the stimuli. But as the authors explain themselves in the reply to my previous concern, if a group of children do not look longer at one of the stimuli, then one cannot draw any firm conclusion. If they found this for bilingual children (as indicated in the pilot), then one cannot conclude that the bilingual children did not see that one of the stimuli changed. This also means that an investigation into the timing of when a group looks longer at one stimulus compared to the other stimulus is not a reliable indication of when the change was noticed. Bilinguals might keep comparing the stimuli despite or maybe even because of noticing the change, while monolinguals (who are more interested in building up detailed representations) might stop comparing.

In conclusion, I think that more explorative behaviour versus more focussed looks to a particular stimulus can tell us something about the different behaviour of monolingual and bilingual infants, but it might not tell us anything about whether the infants noticed the change (in the case that the looks are not longer to one of the two stimuli) or whether one group of infants notices the change earlier.

Converging evidence (e.g., linking behavioural and event-related potential (ERP) measures of infant attention and memory; Reynolds, Courage, & Richards, 2010) support the decades-long held assumption that if preverbal infants are presented with two or more visual stimuli, it is possible to establish systematic visual preferences among them (Chase, 1937; Fantz, 1958, 1964; for review, see Colombo & Mitchell, 2009). However, we agree that theoretically it is possible that infants will continue to look equally at both stimuli even if they notice a change in one – so we will only be able to conclude that these infants did not show a pattern of looking consistent with noticing the change.

Minor points:

pilot data for experiment 4: it is not clear whether the monolingual children looked more to the familiar of changing stimulus.

The monolingual children looked more at the familiar stimulus. Although infant habituation paradigms and similar procedures (like Experiment 4) assume that infants prefer looking at novel stimuli, in reality, fully familiarized infants prefer novel stimuli but not-fully-familiarized infants prefer familiar stimuli (Hunter & Ames, 1998; Hunter, Ross, & Ames, 1982; Roder, Bushnell, & Sasseville, 2000;

Rose, Gottfried, Mello-Carmina, & Bridger, 1982). Although this may make some experiments uninterpretable, for many experimental designs (including Experiment 4) any systematic preference is interpretable – either a familiarity preference or a novelty preference (see Oakes, 2010, for discussion). As long as a preference (either for the novel or for the familiar) is shown, it would mean that the infants discriminated between the two stimuli. If the infants do not show a preference, then it can be concluded that the infants do not show a pattern of looking consistent with discriminating between the two (Colombo & Mitchell, 2009; Oakes, 2010).

p. 11, line 27: do the authors mean they will test whether monolinguals will look longer at the familiar stimulus than predicted by chance (not the novel one)?

We thought it would be more intuitive for readers if we were to compare proportion of looking to the novel AOI with chance level. But it would not matter whether we use proportion of looking to the novel AOI or proportion of looking to the familiar AOI – as long as we compare proportion of looking to one of the AOIs against chance level (.50). For example, if the infants show a familiarity preference by looking at the familiar stimulus for 4 seconds and the novel stimulus for 1 second, then it doesn't matter whether we compare chance (.50) to proportion of looking to the familiar AOI (.80) or to proportion of looking to the novel AOI (.20).

Could the authors please add a note in the text about why the background music was used in experiment 4 (they explained this in the response to my question, but not in the document)

We have now done this.

Kind regards,
Dean D'Souza (and on behalf of the co-authors)

References

- Chase, W. P. (1937). Color vision in infants. *Journal of Experimental Psychology*, 20, 203-222.
- Colombo, J., & Mitchell, D. W. (2009). Infant visual habituation. *Neurobiology of Learning and Memory*, 92(2), 225-234.
- Fantz, R. L. (1964). Visual experience in infants: Decreased attention familiar patterns relative to novel ones. *Science*, 146, 668-670.
- Fantz, R. L. (1958). Pattern vision in young infants. *Psychological Record*, 8, 43-47.
- Hunter, M. A., & Ames, E. W. A multifactor model of infant preferences for novel and familiar stimuli. In: Lipsitt LP, editor. *Advances in child development and behavior*. New York: Academic; 1988. pp. 69–95.
- Hunter, M. A., Ross, H. S., & Ames, E. W. (1982). Preferences for familiar or novel toys: Effects of familiarization time in 1-year-olds. *Developmental Psychology*, 18, 519-529.

- Oakes, L. M. (2010). Using habituation of looking time to assess mental processes in infancy. *Journal of Cognition and Development, 11*(3), 255-268.
- Reynolds, G. D., Courage, M. L., & Richards, J. E. (2010). Infant attention and visual preferences: Converging evidence from behavior, event-related potentials, and cortical source localization. *Developmental Psychology, 46*(4), 886.
- Roder, B. J., Bushnell, E. W., & Sasseville, A. M. (2000). Infants' preferences for familiarity and novelty during the course of visual processing. *Infancy, 1*, 491-507.
- Rose, S. A., Gottfried, A. W., Mello-Carmina, P., & Bridger, W. H. (1982). Familiarity and novelty preferences in infant recognition memory: Implications for information processing. *Developmental Psychology, 18*, 704-713.

Appendix C

Royal Society Open Science Registered Report Stage 2

Dear Sir/Madam,

We have pleasure in submitting our Stage 2 report entitled:

Is mere exposure enough? The effects of bilingual environments on infant cognitive development

Neither the manuscript nor any parts of its contents have been or are currently under consideration or published in another journal.

Page 23 of the Stage 2 manuscript contains the URL for archived study data, digital materials/code, and the laboratory log. Page 23 also contains the URL for the approved Stage 1 protocol on the Open Science Framework.

We can confirm that the completed experiments were executed and analysed in the manner originally approved. No data for any pre-registered study other than pilot data included at Stage 1 was collected prior to the date of IPA. In the Introduction, we would like to correct a minor grammatical error, from "...explain when, how, and why learning two or more languages improve cognitive control" to "...explain when, how, and why learning two or more languages improves cognitive control". Minor changes in section 3.2.3. of the Methods were approved by the editor (e.g., the central fixation stimulus is actually 2.6° wide, not 3.1° wide). Tracked changes were used throughout, so all changes pop out. The stated hypotheses have not been amended or appended.

We hope you will find this report informative and interesting.

Yours faithfully,

Dr Dean D'Souza
Senior Lecturer in Psychology
Faculty of Science & Engineering
Anglia Ruskin University
Cambridge CB1 1PT

Appendix D

Royal Society Open Science – Manuscript ID RSOS-180191.R1

Dear Professor Chris Chambers,

We are pleased to hear that our Stage 2 Registered Report RSOS-180191.R1 entitled "Is mere exposure enough? The effects of bilingual environments on infant cognitive development" was recommended for publication in *Royal Society Open Science*, and that the reviewers suggested only minor revisions. Please see below for our responses (in green) to the reviewers' and your comments.

Associate Editor (Professor Chris Chambers):

Two of the expert reviewers who approved the manuscript at Stage 1 have now assessed the Stage 2 submission. Both reviews are positive overall, with the Stage 2 criteria largely met. There are, however, some constructive comments to attend to in revision. Reviewer 1 makes a useful suggestion regarding the Discussion and ensuring that the conclusions are appropriate given the evidence. Reviewer 3 makes an interesting point about the definition of anticipatory looks and implies a possible reanalysis using different parameters. This analysis is not necessary to achieve Stage 2 acceptance, as these design characteristics were assessed and approved at Stage 1. However, the authors are welcome to perform additional (transparently exploratory) analyses using different criteria, and should at a minimum respond to this point in the response to reviewers (and possibly the Discussion). Reviewer 3 also suggests some restructuring to present the results immediately after each experiment. This seems sensible provided it does not lead to any unnecessary changes to the approved Stage 1 component of the manuscript, but again is not required and I will leave this to up to the judgment of the authors. Please do switch future tense to past tense as the reviewer requests.

We reanalysed the anticipatory looks using a different definition and parameters, but observe a similar pattern (see Additional Analyses below).

We also revised the Discussion in response to the reviewers' suggestions, ensured that all conclusions are appropriate, and switched future tense to past tense.

While we were revising the Discussion, it occurred to us that we were making an implicit assumption: that the ability to disengage attention in order to switch attention (experiment 3) is related to frequency of switching attention (experiment 4) in bilingual (but not necessarily monolingual) infants. This raises the question of whether the two measures actually are related. It makes sense to assume that they are – but it's possible that they are independent. So, we added a section called "4.5.3. Exploratory analyses" and tested the implied relationship. Interestingly, we found that we do indeed observe a relationship between the experiment 3 measure and the experiment 4 measure – in bilingual infants ($r_s = -.40$, $p = .023$) but not in monolingual infants ($r_s = -.02$, $p = .915$). We have added this important (but overlooked) step to the manuscript – but stress in the Results and Conclusion/Discussion that this analysis is "exploratory".

Reviewer 1:

I have read over this Stage 2 registered report and am satisfied that it meets the standards set out in the Stage 1 report.

I did not see code or materials in the linked OSF repository, only lab logs and preregistration documents. There was also no Dryad link. I thus could not verify that these files were shared.

The code was uploaded with the data files to Dryad (doi:10.5061/dryad.3n5tb2rc6).

To download the code (and data) from Dryad, please click on the following temporary link:

https://datadryad.org/stash/share/2BtAvnaEKnaV_VrfF3VfG4Ijn-BX_dUVF8xnc6qc3uw

The temporary link will automatically download all the code and data files.

The code was also uploaded with the data and stimuli to OSF, but we did not want to make that section public until our paper has been published. But to make things easier, we have now made that section public – so hopefully everything is now accessible.

As the authors write in their lab log, "Experimental tasks 2 and 4 were too difficult for the infants and need to be completely redesigned for this age group." This is a real risk of the RR format - the experiments have limited evidential value for reasons that were difficult for both the authors and the reviewers to foresee. Despite this, I found the authors' discussion of the issues generally appropriate.

I would recommend some minor revision of the discussion section. First, the initial paragraph should be broken up into several distinct paragraphs with different points - at least a clear summary of the current finding and a discussion with respect to related work. Second, the oblique "data from one lab" comment felt odd here because the finding that the authors fail to replicate is from Kovacs & Mehler, not (what I assume to be) the Bialystok group.

We have broken up the initial paragraph. For clarity, we have restructured the Discussion. We now start with a very short separate section entitled 'Conclusion'. This follows on from the results and is a clear summary of the current findings. We then continue with a much longer 'Discussion' with respect to related work.

We have deleted the "data from one lab" comment.

Minor:

- "This demonstrates the importance of registered reports" - I don't think this line is warranted in the conclusion. RRs do not guarantee comparable procedures or analyses across groups.

We have deleted the comment.

Reviewer 3:

1. Are the data able to test the authors' proposed hypotheses by passing the approved outcome-neutral criteria (such as absence of floor and ceiling effects or success of positive controls)?

The data analysis follows what had been proposed. In addition, some useful explorative analyses were conducted. Those make a lot of sense and are doing justice to the data.

Having said this, I have a comment on the analyses of Experiments 1 and 2. I noticed that anticipatory looks are defined as saccades within 1s starting 150ms after cue offset. It seems that this is what Kovacs and Mehler (2009) did. But I find the start of this period quite restricted. For instance, in Experiment 1, the last cue shape is presented for 0.8s. Isn't it possible that infants move their eyes before that last cue shape disappears? In other words, would an earlier start of that period make a difference to the results? I would have thought 150ms after the onset (not offset) of the last cue would be more appropriate.

We have now carried out these additional analyses (see the analyses section at the bottom of this page), but the pattern is the same. For example, in Experiment 1, there is some (albeit limited) evidence that the bilingual infants learned to anticipate the reward more quickly than the monolingual infants, which would fit with our proposal, but no evidence that *only* bilinguals can learn to inhibit a learned response. In other words, in the crucial post-switch trials, both groups demonstrated a similar rate of learning.

Defining a single window for all infants/trials is difficult. Because we wanted to replicate Kovacs and Mehler, we stuck to their parameters - but we agree that a different window may have been more appropriate. We had already carried out one exploratory analysis (section 4.2) which only included looks captured within the 1000 ms anticipatory period – the time between last cue offset and target onset. The results were similar to the results from the planned analysis in RR Stage 1.

To check whether the infants made anticipatory looks during the last cue, we extracted looking data from 150ms after the onset of the last cue to the offset of the last cue. For the vast majority of trials, most infants focussed on the cue; we observed very few anticipatory looks. Also, if we define the window as 150 ms after the onset of the last cue to 150 ms after the onset of the target, then the window would be 1800 ms long. We worry that this would capture a lot of noise, which would make it difficult to extract the signal. So, we wanted to keep the window as short as possible. (We could have analysed “first look”, but then that would raise concerns over which look should be considered the “first” one.) In addition, it might be possible for an infant to require less than 150 ms to use visual information to initiate a saccade (Canfield, Smith, Brezsnyak, & Snow, 1997). So, for the additional analysis (see below), for Experiment 1, we analysed a 1650 ms window beginning 150 ms after the onset of the last cue and ending at the onset of the target; for Experiment 2, we chose a 3000 ms window, starting 500 ms after cue onset and ending at the onset of the target. We report these analyses below.

We have not included these additional analyses in the RR Stage 2 manuscript because we didn't want to fill the manuscript with too many unplanned, exploratory analyses.

2. Are the introduction, rationale and stated hypotheses the same as the approved Stage 1 submission?

Yes, these are the same.

3. Did the authors adhere precisely to the registered experimental procedures?

Yes, they generally did, apart from some very small changes that the authors have marked in red. But these changes do not change the hypotheses or the conclusions.

To note is that procedure and analysis sections are written in future tense. I am not familiar with the policy of the journal, but I would suggest to change them to past tense.

We have now changed the tense from future to past.

I also would find it helpful if the results of each of the four experiments could follow directly the method section. It is otherwise difficult to remember the details of the experiments.

We agree that the text would flow better if the results directly followed the methods. However, because some readers are known to first skim the results section before deciding on whether to read the entire paper, we want to keep the results in one section, so if the reader wants to quickly locate them then they can.

I would also move the comparison of the socioeconomic status and age of the two participant groups to the participant section.

We agree and have now moved the comparison to the participant section.

4. Where applicable, are any unregistered exploratory statistical analyses justified, methodologically sound, and informative?

The additional analyses are very useful and appear sound to me.

5. Whether the authors' conclusions are justified given the data

I generally agree with the conclusions. But I have some further questions and suggestions.

First, it is not entirely clear how all the results are interpreted. For that it would be useful to start the discussion section with a summary of the findings and then make sure that all findings are discussed.

We now start with a separate summary of the findings. We now continue with a longer discussion of the findings.

Results of experiment 4 (p. 21):

I don't think that the results of the analysis of the proportion of looks to the changing stimuli necessarily mean that the task was too difficult. It could be that infants notice the change, but lose interest. It would therefore be useful to see a graph for this analysis. (see also comment below about graphs for non-significant results). Btw, I do not understand the comment 'so we cannot conclude that the difference is real' (line 53). Which difference do the authors refer to? The difference of length of looks at the familiar objects? I don't think this analysis diminishes this outcome.

We feel that if the infants noticed a change but lost interest, we would see a change from the time they noticed the change to the end of the experiment – but there didn't seem to be any effect of trial at all. If there was, it was too small to detect or assert with confidence.

To avoid confusion, we have changed the sentence from "...the addition of 'group' did not improve the model, so we cannot conclude that the difference is real" to "...the addition of 'group' did not improve the model, so we cannot draw firm conclusions from this analysis".

It intrigues us that—exactly as we predicted—the monolingual infants looked longer at the familiar stimulus than the bilingual infants. However, this could have been due to chance. We do not believe we can draw any firm conclusions from this analysis, and don't want to mislead the reader, so we just report the non-significant results. Our hope is that we can follow this up by testing older children and then we can combine the two datasets (the current one and any new one) into a cross-sectional study.

Relation of findings to similar studies:

While there is a comparison of the Kovacs and Mehler (2009) study and other studies (including the present one), I find this unsatisfying. What exactly is different between the current study and the one(s) that find different results? And could one analyse the current study in a way more closely to what these other studies did? For instance, is the method of the current study exactly the same as that in Kovacs and Mehler (2009), apart from the sample size and the 'engaging stimuli'? (Btw, does the latter refer to the cue or the reward stimulus). Also, is the coding exactly the same?

There were no differences – apart from the ones we discussed. We analysed the data the same way. The design was also exactly the same. We have heard in conferences that there have been other failed replications (including a small preregistered study that has since been extended into a larger, still ongoing registered report). With more replications and bigger sample sizes, we believe that the field will have more clarity.

The latter refers to the reward because we used the exact same cues (the coloured shapes) as Kovacs and Mehler.

Minor comments:

- Please indicate in the figure captions which experiment they belong to.

We have now added these.

- Results of non-significant findings (e.g. Experiment 2): a graph showing the results would be very helpful for the reader.

We had some debate about this. We agree it is usually helpful to plot data. But in our experience, too many readers look at graphs without reading the accompanying text – and often overinterpret what they see. We want to avoid this possibility and just present visualizations of significant results. One of our students would like to run the experiment on older children. We could provide them with the infant data, so they can turn their study into a cross-sectional one.

- P. 10, line 29: change “periphe ral” to “peripheral”

We’re not sure why there is a space in the pdf version of our word document. We thank the reviewer for spotting it and will check that it does not reoccur when the new proofs are generated.

- First paragraph of discussion (p. 22) is too long. Split up into smaller meaningful units.

We have now broken up the first paragraph of the discussion and made it clearer.

We thank the reviewers for their comments and believe that our paper is now much stronger.

Yours sincerely,

Dean D’Souza

Additional analyses

Experiment 1

For the pre-switch trials, we fitted a mixed effects logistic model (the ‘null’ model) with ‘correct anticipatory look’ as the outcome variable and random intercepts for participants. An ‘anticipatory look’ was defined as looking towards the area of interest that occurs from 150 ms after last cue onset to target onset. We assumed random intercepts for participants because it is likely that baseline ‘anticipatory looking’ varies across infants irrespective of group. We then added a fixed effect of trial (the ‘+ trial’ model):

Model (null): correct anticipatory look $\sim (1 \mid \text{participant}) + \epsilon$

Model (+ trial): correct anticipatory look $\sim (1 \mid \text{participant}) + \text{trial} + \epsilon$

If infants learned to anticipate the appearance of the reward, then the addition of ‘trial’ should significantly improve the ‘null’ model. Comparisons of AIC and BIC, as well as a likelihood ratio test, show a highly significant effect of trial (Table A). This suggests that the task was working (see Table B for the estimated fixed effect of trial).

Table A. The relationship between pre-switch trials and correct anticipatory looks.

Model	Df	AIC	BIC	logLik	deviance	χ^2	χ^2 Df	p
Null	2	1185.1	1194.7	-590.54	1181.1			

+ trial	3	1154.0	1168.4	-574.00	1148.0	33.10	1	<.001***
+ group	4	1155.6	1174.7	-573.79	1147.6	0.41	1	.521
Interaction	5	1150.4	1174.4	-570.21	1140.4	7.16	1	.007**

Note: The 'interaction' model fit the data better than the '+ trial' model, $\chi^2(2) = 7.57, p = .023^*$.
 $^*p < .05, ^**p < .01, ^***p < .001$

Table B. Estimated fixed effects (Model: + trial).

Effect	Estimate	Std. Error	Z
Intercept	-1.04	0.19	-5.48
Trial	0.17	0.03	5.64

Adding a fixed effect of group did not improve the model (Table A; see also Figure A1). However, a comparison of AIC and a likelihood ratio test suggest that adding the possibility of an interaction between group and trial (the 'interaction' model) does improve the '+ group' model ($p = .007$). Moreover, the 'interaction' model fits the data better than the (no-group) '+ trial' model (Table A).

Figure A1. Anticipatory looks increased across trials in the pre-switch phase in both groups, indicating learning, more so in bilingual infants than in monolingual infants. **A2.** Anticipatory looks increased across trials in the post-switch phase in both groups, indicating that both groups were successfully redirecting their anticipatory looks.

It is important to note that a comparison of BIC does not favour the 'interaction' model over the '+ trial' model. This may be because BIC penalises additional parameters more heavily than AIC. On the one hand, the '+ trial' model is more parsimonious than the 'interaction' model. On the other hand, the reality may be complex and require additional parameters. In any case, we advise caution when interpreting these results.

To ascertain whether the bilinguals but not monolinguals inhibited the learned behaviour, we fitted a linear mixed effects model (the 'null' model) to the post-switch trials data, with 'correct anticipatory look' as the outcome variable and random intercepts for participants. We then added a fixed effect of trial (the '+ trial' model). Comparisons of AIC and BIC, as well as a likelihood ratio test, show a highly significant effect of trial (Table C). The addition of a

fixed effect of ‘group’ also improved the model (Table C). But importantly, the addition of an interaction did not improve the model (Table C). In short, our data do not support the claim that *only* bilingual infants can inhibit learned behaviour; proportion of correct anticipatory looks increased in both groups.

Table 2. The relationship between post-switch trials and correct anticipatory looks.

Model	Df	AIC	BIC	logLik	deviance	χ^2	χ^2 Df	p
Null	2	1092.1	1101.5	-544.03	1088.1			
+ trial	3	1081.2	1095.5	-537.62	1075.2	12.82	1	<.001***
+ group	4	1076.8	1095.8	-534.42	1068.8	6.39	1	.011*
Interaction	5	1078.8	1102.6	-534.42	1068.8	<0.01	1	.946

$\sim p < .10$, * $p < .05$, ** $p < .01$, *** $p < .001$

Experiment 2

To check that the infants learned to associate a cue (elephant, snowman) with the location of a reward, we fitted a mixed effects logistic model (the ‘null’ model) with ‘correct anticipatory look’ as the outcome variable and random intercepts for participants. We then added a fixed effect of trial (the ‘+ trial’ model):

Model (null): correct anticipatory look $\sim (1 \mid \text{participant}) + \varepsilon$

Model (+ trial): correct anticipatory look $\sim (1 \mid \text{participant}) + \text{trial} + \varepsilon$

If infants learned to anticipate the appearance of the reward, then the addition of ‘trial’ should significantly improve the ‘null’ model. However, comparisons of AIC and BIC, as well as a likelihood ratio test, show that this was not the case (Table D; see also Figure B). This suggests that the task was too difficult for the infants. (For sake of completeness, we added a fixed effect of group, but this did not improve the model. See Table D.)

Table D. The relationship between trials and correct anticipatory looks.

Model	Df	AIC	BIC	logLik	deviance	χ^2	χ^2 Df	p
Null	2	3517.1	3528.9	-1756.5	3513.1			
+ trial	3	3518.6	3536.3	-1756.3	3512.6	0.52	1	.471
+ group	4	3520.2	3543.8	-1756.1	3512.2	0.42	1	.518

Figure B. Anticipatory looks did not increase across trials. We cannot therefore claim that the task was working. The task was, arguably, too difficult. The infants were more likely to make either an incorrect anticipatory look or no anticipatory look.